# Unraveling Arctic submicron organic aerosol sources: a year-long study by H-NMR and AMS in Ny-Ålesund, Svalbard

Marco Paglione<sup>1,†</sup>, Yufang Hao<sup>2,†</sup>, Stefano Decesari<sup>1</sup>, Mara Russo<sup>1</sup>, Karam Mansour<sup>1\*</sup>, Mauro Mazzola<sup>3</sup>, Diego Fellin<sup>4,5</sup>, Andrea Mazzanti<sup>6,7</sup>, Emilio Tagliavini<sup>6</sup>, Manousos Ioannis Manousakas<sup>2,8</sup>, Evangelia Diapouli<sup>8</sup> Elena Barbaro<sup>4,5</sup>, Matteo Feltracco<sup>4,5</sup>, Kaspar R. Daellenbach<sup>2</sup>, Matteo Rinaldi<sup>1</sup>

20 Correspondence to: Matteo Rinaldi (m.rinaldi@isac.cnr.it) and Kaspar R. Daellenbach (kaspar.daellenbach@psi.ch)

importance for particle climate-relevant properties. This study presents a year-long analysis (May 2019 - June 2020) of PMI filter samples collected in Ny-Ålesund, Svalbard. A multi-instrumental approach is employed to characterize the comprehensive chemical composition of PMI, with a specific focus on its water-soluble organic fraction depicted combining proton nuclear magnetic resonance spectroscopy (H-NMR) and high-resolution time-of-flight aerosol mass spectrometry (HR-TOF-AMS), which provide complementary insights on nature and structure of the organic aerosol classes characterizing the bulk OA mixture. Positive Matrix Factorization (PMF) source apportionment identifies consistent OA sources from the H-NMR and AMS datasets, showing a pronounced seasonality in their relative contributions to total OA mass. Winter-spring aerosol is dominated by long-range transport of Eurasian anthropogenic pollution (up to 70%), while summer is characterized by biogenic aerosols from marine sources (up to 44%), including sulfur compounds, amines, and fatty acids. Occasional summertime high OA loadings are associated with wildfire aerosols enriched in levoglucosan and humic-like substances (HULIS; averagely 27-28%). Eventually, about 28-40% of the OA mass is attributed to an unresolved mixture of extremely oxidized compounds of difficult specific source-attribution. This integrated approach provides valuable insights into the seasonal dynamics of OA sources in the Arctic.

Abstract. Understanding the chemical composition and sources of organic aerosol (OA) in the Arctic is critical given its

<sup>&</sup>lt;sup>1</sup>National Research Council of Italy - Institute of Atmospheric Sciences and Climate (CNR-ISAC), Bologna, 40129 Italy

<sup>&</sup>lt;sup>2</sup>Laboratory of Atmospheric Chemistry, Paul Scherrer Institute, Villigen PSI 5232, Switzerland

<sup>&</sup>lt;sup>3</sup>National Research Council of Italy - Institute of Polar Sciences (ISP-CNR), Bologna, 40129, Italy

<sup>&</sup>lt;sup>4</sup>National Research Council of Italy - Institute of Polar Sciences (ISP-CNR), Bologna, 40129, Italy

<sup>50</sup> Department of Environmental Sciences, Informatics and Statistics, Ca' Foscari University of Venice, Venice Mestre, 30172, Italy

<sup>&</sup>lt;sup>6</sup>Department of Chemistry, University of Bologna, Bologna, 40126, Italy

<sup>&</sup>lt;sup>7</sup>Department of Ind. Chemistry, University of Bologna, Bologna, 40136, Italy

 <sup>\*</sup>Environmental Radioactivity & Aerosol Tech. for Atmospheric & Climate Impacts, INRaSTES, National Centre of Scientific
 Research "Demokritos", Ag. Paraskevi, 15310, Greece

<sup>†</sup>these authors contributed equally.

<sup>\*</sup>also at: Oceanography Department, Faculty of Science, Alexandria University, Alexandria 21500, Egypt.

#### 1 Introduction

Atmospheric aerosol exerts potentially significant, yet uncertain effects on the climate system by modulating the radiative balance of the atmosphere and by altering the surface albedo of the cryosphere (IPCC, 2021). Investigating aerosol-climate interactions in the Arctic regions is of paramount importance, given the faster-than-global warming rate of this area, known as Arctic amplification, the multiple atmosphere-ocean-cryosphere feedbacks involved, and the proximity to strong pollution sources in the northern hemisphere mid-latitudes (Serreze and Barry, 2011; Schmale et al., 2021).

Organic aerosol (OA) is one of the most abundant fraction of fine aerosol mass globally and also in the Arctic. However, in spite of its fundamental role in modulating climate-relevant properties of airborne particles, OA chemical composition and

spite of its fundamental role in modulating climate-relevant properties of airborne particles, OA chemical composition and sources are still poorly understood in the polar regions, mainly due to the measurement difficulties in harsh environments and the consequent scarcity of long-term observational datasets with sufficient chemical detail (Schmale et al., 2021; AMAP 2021). Long-range atmospheric transport (LRT) of air masses from lower latitudes is recognized as an important driver of the Arctic aerosol burden, since local emissions of submicron particles are relatively much smaller (e.g., Quinn et al., 2007). However, thirty years of monitoring at Arctic observatories have shown that the contribution of LRT to Arctic aerosol concentrations is consistently decreasing, as a consequence of reduced emissions in the mid-latitudes (Collaud Coen et al., 2020; AMAP 2021; Schmale et al., 2022). In turn, local pollution from resource extraction and shipping (e.g., Peters et al., 2011; Pizzolato et al., 2016) as well as, most importantly, natural aerosol sources may become increasingly significant in the future. For instance,

wildfires (WFs) increased (McCarty et al., 2021), as well as sea salt aerosol (SSA) (Heslin-Rees et al., 2020), and eolian mineral dust (MD) linked to glacial retreat (Groot Zwaaftink et al., 2016). Additionally, primary biological aerosol particles (PBAPs) are expected to increase due to permafrost thawing and Arctic greening (Myers-Smith et al., 2020), which may also enhance the emission of biogenic volatile organic compounds, leading to higher levels of biogenic secondary organic aerosol (BSOA) (Hallquist et al., 2009). These shifts in sources could further modify the physicochemical properties of Arctic aerosols and consequently their climate impact.

Gaining insights into changes in Arctic aerosol emissions and formation, as well as variations in LRT, and aerosol chemical composition is essential for assessing their effects on climate and the changing Arctic environment. In this regard, long-term observations are becoming increasingly fundamental.

The lack of systematic observations of OA composition is among the critical knowledge gaps in aerosol monitoring in the Arctic (AMAP 2021). The available studies (Nielsen et al. 2019; Frossard et al., 2011; Chang et al., 2011; Leaitch et al., 2018) are single-site, short-term or campaign-based, and too infrequent to reveal seasonal patterns in the main sources of Arctic OAs. Hence, Arctic aerosol source apportionment studies typically do not consider OAs (Polissar et al., 1998; Nguyen et al., 2013) or are limited to a few primary markers (for example, levoglucosan from biomass burning or elemental carbon - EC), or again focus on radiocarbon measurements (Winiger et al., 2019; Rodriguez et al., 2020). Therefore, current efforts have so far been unable to provide an understanding of the sources and formation pathways of the pan-Arctic OAs in different seasons.

While organic carbon (OC) has been continuously monitored in Alert (Canada) since 2006 and in Ny-Ålesund (Svalbard) since 2011 (AMAP 2021), information on low-molecular weight organic acids was first obtained during a nearly 2-year-long study at Sevettjärvi in the lower Arctic (Finland) (Ricard et al., 2002). Later studies in Ny-Ålesund have focused on levoglucosan, sugars, methansulfonate (MSA) and biogenic secondary organic aerosols (BSOA) tracers (Yttri et al., 2014, 2024; Karl et al., 2019; Becagli et al., 2019; Gramlich et al., 2024), whereas Moschos et al. (2022) presented the most comprehensive study on Arctic OA based on aerosol mass spectrometry (AMS) to this date, with up to 3 years of data from eight Arctic sites.
In particular, Moschos et al. (2022) quantified the sources of Arctic organic aerosol (OA) by analyzing its water-soluble

In particular, Moschos et al. (2022) quantified the sources of Arctic organic aerosol (OA) by analyzing its water-soluble fraction using Aerosol Mass Spectrometry (AMS) on samples collected across the Arctic, followed by the application of positive matrix factorization (PMF). Their study identified six aerosol factors, three primarly linked to anthropogenic sources (namely, Arctic haze, primary organic aerosol, and oxygenated organic aerosol) and three associated with natural emissions (primary biological organic aerosol, methane sulfonic-acid-related organic aerosol, and biogenic secondary organic aerosol). These factors displayed distinct seasonal patterns, with the anthropogenic sources prevailing in winter, while those of natural origin more prominent in summer.

While this offline AMS approach provides chemical fingerprints of WSOA, it only detects fragments of the original molecules because of the hard ionization pathways in the instrument. H-NMR offers information on the organics functional groups, complementary to the mass spectral fingerprints from the AMS. Compared to AMS, H-NMR spectroscopy exhibits inferior selectivity to organic compounds oxidation state, while it provides a better split between aromatic and aliphatic groups and retains source-specific information in the distribution of aliphatic H-NMR resonances (Decesari et al., 2024). The availability of more complete spectral databases for atmospherically-relevant compounds and the employ of factor analysis techniques have greatly improved the potential of this technique for organic source apportionment. Most importantly, modern H-NMR techniques have achieved substantial gain in sensitivity with respect to the first atmospheric studies, enabling application to aerosol characterization in remote environments (Tagliavini et al., 2024). Recent exemplar investigations in Antarctica have focused on primary and secondary natural sources (Decesari et al., 2020; Paglione et al., 2024). Here, we present the first yearlong investigation combining AMS, H-NMR, ion chromatography (IC), organic and elemental carbon (OC and EC, respectively) measurements to characterize OA in the high Arctic. Conducted at the Gruvebadet Laboratory in Ny-Ålesund,

# 2 Material and methods

# 2.1 Measurement field campaign

05 The measurements reported here were done in the framework of the Joint Research Center ENI-CNR-Aldo Pontremoli, within the ENI-CNR Joint Research Agreement (Donateo et al., 2023), and of the NASCENT (Ny-Ålesund Aerosol Cloud

Svalbard, this study provides seasonal insights into anthropogenic and biogenic contributions to Arctic OA.

Experiment, Pasquier et al., 2022) study in the period May 2019 - June 2020 (Figure S1), exploiting the facilities of the Ny-115 Ålesund Research Station in Svalbard (Norway). More specifically the sampling took place at the Gruvebadet Laboratory (GVB) located southwest of the village of Ny-Ålesund (78° 55' N, 11° 56' E, Figure 1).

A high-volume sampler (TECORA ECHO HiVol, equipped with Digitel PM<sub>1</sub> sampling inlet) collected ambient aerosol particles with Dp<1 µm on pre-washed (with 250 mL of ultrapure Milli-Q water) and pre-baked (1h at 800° C) quartz fiber filters, at a controlled flow of 500 L min<sup>-1</sup>. The sampler was located at GVB, and the sampling head was at about 4 m above the ground, easily accessible on the roof of the building. Due to the necessity of collecting sufficient aerosol loading for the detailed chemical analyses on the filters, the sampling time was of the order of about 4 days (84 ± 9 h, on average ± standard deviation) for each sample. A total of 87 PM<sub>1</sub> samples and 5 field blanks (same quartz-fiber filters mounted on the sampler but not sampled) were collected along the study period. After sampling the filters were stored in a freezer and then shipped in thermal insulated boxes to the CNR-ISAC laboratory in Bologna, Italy, where they were kept frozen at about -20° C until extraction and chemical analyses. About half of each filter was used for the off-line characterization of water-soluble organic carbon (WSOC) by Total Organic Carbon (TOC) analyzer, Ion-chromatography and H-NMR (as described below) performed in Bologna, while a quarter was shipped to PSI laboratories in Zurich, Switzerland for AMS measurements and carbonaceous content quantifications (both WSOC by TOC analyzer and EC/OC on filters by Sunset, as detailed below).

2.2 Aerosol offline chemical characterization

To determine organic carbon (OC) and elemental carbon (EC), the Thermo-Optical Transmittance (TOT) method was employed using a Lab OC-EC Aerosol Analyzer (Model 5L, Sunset Laboratory Inc., USA) following the EUSAAR2 protocol. Quartz filter samples were subjected to controlled heating according to the EUSAAR2 thermal protocol: initially, they were heated up to 650 °C in an inert helium (He) atmosphere to evolve OC. Subsequently, they were heated to 850 °C in a mixture of 2% oxygen in He, where EC was oxidized. The continuous monitoring of the sample's transmittance during the heating process allowed charring correction application, as detailed in Popovicheva et al. (2024). The method's limit of detection (LOD) was 0.02 µg/m<sup>3</sup> of carbon. To ensure consistency, field and laboratory blanks were processed using the same procedures. The expanded uncertainties for the analysis were calculated to be 15% for OC and 23% for EC.

The aerosol filter samples were extracted with deionized ultrapure (Milli-Q) water using a mechanical shaker for 1 h. In order to remove suspended materials the water extracts were filtered on PTFE membranes (pore size: 0.45 µm). Extracts were analyzed by means of a TOC thermal combustion analyzer (Shimadzu TOC-5000A) for the quantification of WSOC content. Ion chromatography was applied to the extracts for the quantification of the main water-soluble inorganic ions (sodium, Na+; chloride, Cl<sup>+</sup>; nitrate, NO<sub>3</sub><sup>-</sup>; sulfate, SO<sub>4</sub><sup>2-</sup>; ammonium, NH<sub>4</sub><sup>+</sup>; potassium, K<sup>+</sup>; magnesium, Mg<sup>2+</sup>; calcium, Ca<sup>2+</sup>), some organic acids (acetate, ace; formate, for; methanesulphonate, MSA; oxalate, oxa) (Sandrini et al., 2016) and low molecular weight alkyl-amines (methyl-, ethyl-, dimethyl-, diethyl- and trimethylamine, MA, EA, DMA, DEA and TMA, respectively) (Facchini Eliminato: the Gruvebadet laboratory

Fliminato:

155 et al., 2008a). An IonPac CS16 3 × 250 mm Dionex separation column with gradient MSA elution and an IonPac AS11 2 × 250 mm Dionex separation column with gradient KOH elution were deployed for cations and anions, respectively. Seasalt and non-seasalt fractions of the main inorganic ions measured by IC (ss-x and nss-x, respectively) were quantified based on the global average sea-salt composition found in Seinfeld and Pandis (2016) using Na+ as the sea-salt tracer. All the data are available at IADC Data repository (https://doi.org/10.71761/0e110925-1f3d-4013-b048-e5a47ca3be6f). Field blanks were collected and all the sample concentrations were corrected for the blanks. Detection limits (LODs, blanks average concentrations + 2x standard deviation of blanks filter concentrations) of each species are reported in Table S1.

# 2.2.1 H-NMR analysis of the samples

Following well-established procedures (Decesari et al., 2000), aliquots of the aerosol water extracts were dried under vacuum and re-dissolved in deuterium oxide (D<sub>2</sub>O) for the H-NMR spectroscopy (hereinafter also referred as NMR) characterization of organic functional group. In order to allow the quantification of the spectral signals, sodium 3-trimethylsilyl- (2,2,3,3-d4) propionate (TSP-d4) was used as an internal standard by adding 50 µL of a 0.05% TSP-d4 (by weight) in D<sub>2</sub>O to the sample in the tube. To avoid the shifting of pH-sensitive signals, the extracts were buffered to pH~3 using a deuterated-formate/formicacid (DCOO = HCOOH) buffer prior to the analysis. The H-NMR spectra were then acquired at 600 MHz in a 5 mm tube using a Varian Unity INOVA spectrometer, at the NMR facility of the Department of Industrial Chemistry of the Bologna University, H-NMR spectroscopy in protic solvents provides the speciation of hydrogen atoms bound to carbon atoms. On the basis of the range of frequency shifts, the signals were attributed to H-C containing specific functionalities (Decesari et al., 2000, 2007). A comprehensive list and description of the functional groups, molecular species and categories of compounds identified by H-NMR spectra analysis in this study is reported in Table S2. Briefly, the main functional groups identified included: unfunctionalized alkyls (H-C), i.e. methyls (CH3), methylenes (CH2), and methynes (CH) groups of unsubstituted aliphatic chains (i.e., also named later "Aliphatic chains", chemical shift range 0.5-1.8 ppm); aliphatic protons adjacent to unsaturated/substituted groups (benzyls and acyls: H-C-C=) and/or heteroatoms (amines, sulfonates: H-C-X, with X≠O), like alkenes (allylic protons), carbonyl or imino groups (heteroallylic protons) or aromatic rings (benzylic protons) (i.e., also named later "Polysubstituted aliphatic chains", chemical shift range 1.8-3.2 ppm); aliphatic hydroxyl/alcoxy groups (H-C-O), typical of a variety of possible compounds, like aliphatic alcohols, polyols, saccharides, ethers, and esters (i.e., also abbreviated later as "Sug-Alc-Eth-Est", chemical shift range 3.2-4.5 ppm); anomeric and vinyl groups (O-CH-O, chemical shift range 5-6.5 ppm), from not completely oxidized isoprene and terpenes derivatives, from products of aromatic-rings opening (e.g., maleic acid), or from sugars/anhydrosugars derivatives (glucose, sucrose, levoglucosan, glucuronic acid, etc.); and finally aromatic functionalities (Ar-H, also abbreviated later as "Arom", chemical shift range 6.5-9 ppm). Organic hydrogen concentrations directly measured by H-NMR were converted to organic carbon using stoichiometric H/C ratios specifically assigned to functional groups using the same rationale described in previous works (Paglione et al., 2024). Although the sum of NMR functional group concentrations approached total WSOC in many samples, the uncharacterized fraction using the typical H/C

ratios was significant (on average 30%, Figure S2). Possible reasons for the "unresolved carbon" are (1) volatiles/semi-volatiles losses during the evaporation of the extract prior to the preparation of the NMR tube, (2) the presence of carbon atoms not protonated, thus not-detectable to H-NMR, like oxalates or such as compounds containing substituted quaternary carbon atoms or fully substituted aryls (Moretti et al., 2008), and (3) the uncorrected estimations of stoichiometric H:C ratios used for the conversion of directed measured organic hydrogens into organic carbon. Considering the reasons (2) and (3) as the most relevant (meaning that the lower recovery mainly reflected a certain abundance of organic moieties not carrying H-C bonds) we re-calculated the H-NMR total WSOC by applying a H/C conversion ratio of 1 to the "Polysubstituted aliphatic chains" to account for carbon in aliphatic carbonyl/carboxyl groups under the assumption that every carbon atom in acyls (H-C-C=) is adjacent to a carbonyl/carboxyl carbon (hence H-C-C=O) (as in Decesari et al., 2007). In this way the recovery of H-NMR analysis approaches the closure with the TOC-analyzer derived concentration of WSOC (slope = 1.01; Figure S2) indicating that carbonyl/carboxylic groups can account for a large fraction of the carbon atoms unbound to hydrogen atoms in these samples.

Some organic tracers were identified in the H-NMR spectra on the basis of their characteristic patterns of resonances and

chemical shifts, using for this scope libraries of reference spectra from the literature (of standard single compounds and/or mixtures from laboratory/chamber experiments and/or from ambient field studies at near-source stations). The identification was confirmed by means of NMR chemical shift elaboration software leveraging extensive libraries of biogenic compounds, such as the Chenomx NMR suite (Chenomx inc., evaluation version 9.0), or based on simulated H-NMR spectra of atmospheric relevant molecules, such as ACD/Labs (Advanced Chemistry Developments inc., version 12.01). Among such tracers, methane-sulfonic acid (MSA, singlet at 2.80 ppm) and low-molecular-weight alkyl amines (namely, di- and tri- methyl amines, DMA and TMA respectively, at 2.72 and 2.89 ppm) were quantified. Speciation and quantification of these tracers by H-NMR were validated by comparison with the IC measurements showing excellent agreements between the two techniques for MSA and a reasonable correlation for TMA and total amines (Figure S3). The quantification of amines was anyway challenging, due to the very low concentrations (especially for IC) and because of possible overlapping signals from other compounds present at similar concentrations in NMR spectra. Other tracers quantified by H-NMR were levoglucosan and hydroxymethane sulfonate (HMS), with their unequivocal signals at 5.45 and 4.39 ppm, respectively (Suzuki et al., 2001; Paglione et al., 2014b) (Figure S4). Additional tracers (such as sucrose, glucose, glycerol, etc.) were identified, but not quantified because

# 2.2.2 AMS measurements and data analysis.

The offline AMS technique established by Daellenbach et al. (2016) was used here. Briefly, punches from the quartz fiber filter samples were extracted in ultrapure water (18.2 Mohm cm, total organic carbon  $\leq$ 3 ppb by weight) and inserted into an ultrasonic bath for 20 min at 30 °C. Typical organic concentrations of the quartz-fiber-filtered water-extracts were 2–3  $\mu$ g C ml<sup>-1</sup>. Each sonicated extract was then filtered through a syringe-filter made of a nylon membrane (0.45  $\mu$ m; Infochroma AG)

and transferred to a 'Greiner' sample tube (50 ml). For a better quantification of the aerosol species, each extract was spiked with a known quantity (6 ppm) of isotopic labeled internal standard of ammonium nitrate (NH4<sup>15</sup>NO3). From the obtained solutions, aerosols were generated in synthetic air (80% volume N2, 20% volume O2; Carbagas) via an apex Q nebulizer (Elemental Scientific, Inc.) operated at 60 °C, dried by a Nafion dryer and directed into a long-time-of-flight AMS. Each sample was recorded for ca. 210 s, with a collection time for each spectrum of ~33 s. Ultrapure water was measured with the same modality before each sample measurement to assess the instrumental background during the analysis of the corresponding sample.

35 For the data analysis, we used Squirrel v1.57I for the m/z calibration and baseline subtraction, and Pika v1.16I for high-resolution (HR) analysis, in the Igor Pro software package 6.37. The HR peak fitting was performed in the m/z range 12–210. After the peak fitting, the atmospheric concentration of each ion was calculated by normalizing to the known <sup>15</sup>N concentration in the sample (from the labeled internal standard) and considering the appropriated Relative Ionization Efficiency (RIE), the extraction volume and portion of the filter that was extracted for analysis. The average water-blank signal was subtracted from the average signal of the following sample and, eventually, all the obtained concentrations were corrected by the average concentrations obtained analyzing the field blanks. Elemental ratios were calculated using the Improved Ambient approach (Canagaratna et al., 2015) from the blank-corrected mass spectra (Figure S5).

## 2.2.3 Water-soluble and water-insoluble organic matter

Using the WSOC and OC measured by elemental analysis (Sect. 2.2) and the Organic Matter-to-Organic Carbon (OM:OC) ratio provided by the HR-ToF-AMS (Sect. 2.2.2) the following aerosol components have been defined:

- (1) Water Soluble Organic Matter (WSOM) = WSOC \* (OM:OC)<sub>AMS</sub>
- (2) Water Insoluble Organic Matter (WIOM) = (OC WSOC) \* 1.4 (Rinaldi et al., 2013)
- (3) Organic Matter (OM) = WSOM + WIOM

# 2.3 Auxiliary measurements

Additional off-line analyses and on-line measurements were carried out and used to corroborate and validate the AMS and H-NMR measurements and source apportionments.

Organic markers were determined in parallel samples collected by an Andersen multi-stage impactor managed by CNR-ISP (Institute of Polar Sciences) every 6-10 days from 16th June 2019 to 2nd January 2020. Polyols and sugars (Barbaro et al., 2015a) and inorganic ions and organic acids (methanesulfonic acid and C2-C7 carboxylic acids) (Barbaro et al., 2017; Feltracco et al., 2021) were determined using ion chromatography coupled with mass spectrometry, while free amino-acids (FAAs) (Barbaro et al., 2015b), phenolic compounds (PCs) (Zangrando et al., 2016) and photo-oxidation products of α-pinene

(Feltracco et al., 2018) were measured by high-performance liquid chromatography (HPLC) coupled with triple quadrupole mass spectrometer.

Evaluations of Equivalent Black Carbon (eBC) were obtained at Gruvebadet through continuous online measurements carried out by means of a Particulate Soot Absorption Photometer (PSAP) (Gilardoni et al., 2020, Gilardoni et al., 2023).

#### 2.4 Source regions classification

The concentration-weighted trajectory (CWT) method is used to assess the potential impacts of long-range aerosol transport (Mansour et al., 2022; Rinaldi et al., 2021). It combines the residence time of air masses (trajectory points) over geographic areas with particle concentrations measured at a specific receptor site. In this study, the CWT was applied to identify the most likely source regions contributing to OA components detected using HR-AMS and H-NMR at Ny-Ålesund. Backward air mass trajectories reaching 100 meters above ground level were calculated using the NOAA HYSPLIT4 (https://ready.arl.noaa.gov/, last accessed on 5 August 2022) model (Rolph et al., 2017; Stein et al., 2015). Despite the general uncertainties of the backtrajectories in the Arctic due to the lack of meteorological measurements to constrain the model (e.g., Harris et al., 2005; Kahl, 1993), this approach is widely used in supporting identification of the source area of the different aerosol components measured at a receptor site. The trajectory calculations were driven by meteorological data from the archived Global Data Assimilation System (GDAS1; 1° × 1°) of the National Centers for Environmental Prediction (NCEP (ftp://arlftp.arlhq.noaa.gov/pub/archives/gdas1, last access: 01 August 2022). The trajectories were traced back over 10 days, with points recorded at 1-hour intervals along each track. Air mass arrival frequency was set to every 6 hours (four trajectories per day) throughout the sampling period from May 2019 to June 2020. For each filter sample, the tracks spanning the sampling time from start to end represent the pathways of incoming air masses. A detailed description of the applied equation and calculation protocols can be found in Rinaldi et al. (2021).

# 280 2.5 Factor analysis of AMS and H-NMR Spectra

In order to better investigate the variability of OA composition and to apportion its sources, we exploit non-negative factor analysis applied separately to the series of AMS and NMR spectra of the WSOM and we then compared the results for a better interpretation and chemical characterization of the resulting factors as organic aerosol sources.

In particular we applied to both the collections of NMR and AMS spectra the "Positive matrix factorization" (PMF, Paatero et al., 1994), using the ME-2 solver (Paatero et al., 1999) implemented within the Source Finder toolkit (SoFi pro version 8.6, Datalystica Ltd) for Igor Pro (WaveMetrics, Inc) (Canonaco et al., 2013). The aim of PMF is to derive a linear combination of components (factors) that can reproduce the observed chemical composition and variations in time of the samples (Zhang et al., 2011). The PMF was applied to NMR and AMS datasets separately, following the method already described in previous publications (Paglione et al., 2014a, 2014b, 2024 for NMR; Daellenbach et al., 2016, 2017; Bozzetti et al., 2017a, 2017b; Casotto et al., 2022; Moschos et al., 2022 for AMS-offline). Briefly, about NMR, the original spectra were subjected to several

preprocessing steps in order to remove spurious sources of variability before the statistical analysis. Baseline subtraction based on a polynomial fit was applied to each spectrum. Careful horizontal alignment of the spectra was performed using the Tsp-d4 and buffer singlets as reference positions (at 0.00ppm and 8.45ppm, respectively). The spectral regions containing only noise or sparse signals of solvent/buffer (H< 0.5 ppm; 4.7 < H< 5.2 ppm; and 8.15< H< 8.60 ppm) were removed. The five NMR spectra of the blanks were averaged together and the corresponding mean blank-spectrum was subtracted to all the sample-spectra. Binning over 0.02 ppm of chemical shift intervals was applied to remove the possible variability of peak position produced by matrix effects. Low-resolution spectra (~400-points) were finally obtained and processed by PMF. The error input matrix required by PMF was derived from the signal-to-noise ratios of the NMR spectra (as already described in previous publications, Paglione et al., 2014a, 2014b and 2024), calculated for each sample as 7 times the standard deviation of the signal intensity in a portion of the spectrum containing only noise/baseline values (between 6.5 and 7ppm). Solutions with up to eight factors were explored.

About AMS, the input matrixes for organic aerosol source apportionment were prepared by eliminating all the isotope ions and the inorganic fragments from the quantified ions. The error matrix was prepared following the error propagation approach by Daellenbach et al., 2016. The average signal-to-noise ratio was >2.0 for 278 out of 389 (71%) fitted organic fragment ions with m/z up to 130 and for 180 out of 315 (57%) for masses above m/z 130.

A full examination of the PMF application (inputs preparation, uncertainty estimations, etc.) and outcomes (range of solutions investigated, procedure for choosing best solutions, analysis of the residuals, model error evaluation, interpretation of the results etc.) on both the NMR and AMS datasets is reported in the Supplementary (Section S2, Figure S5-S19, Table S3), while in section 3.3 we focus on a four-factors and five-factors solutions for AMS and NMR respectively, for which a substantial agreement between the two approaches (i.e., AMS and NMR -based factor analysis) was achieved. Interpretation of factors and their attribution to specific sources was based on an integrated approach including: the comparison between spectral profiles and a unique library of reference spectra (recorded during laboratory studies or in the field at near-source stations (Paglione et al., 2014a; Paglione et al., 2014b; Decesari et al., 2020; Paglione et al., 2024); the correlation of factors contributions with available chemical tracers (i.e., sea salt and other inorganic ions, levoglucosan, eBC, sugars and polyols, free-aminoacids and organic acids, MSA and amines, reported in supplementary Tables S2 and Figure S14); and the examination of backtrajectories and of the concentration-weighted trajectories (CWT) maps of each factor indicating their potential source areas (further discussed below).

# 3 Results

The results section is divided in three main parts as following. In section 3.1 we discuss the variability in the bulk aerosol composition at Ny-Ålesund and its possible drivers in 2019-2020. In section 3.2 we focus on the AMS and NMR spectroscopic

characterization of the organic fraction of the aerosol. Finally, in section 3.3 we discuss the WSOA source apportionment results based on the PMF applied on the AMS and NMR spectral datasets.

## 3.1 Main PM1 chemical composition and seasonality

The chemical composition of PM1 aerosol at Ny-Ålesund during the period May 2019 - June 2020 and its seasonality is summarized in Figure 2 (where summer = June + July + August, fall = September + October + November, winter = December + January + February and spring = March + April + May). Here PM1 total mass is considered the sum of the main water-soluble species measured by ion-chromatography (seasalt, nss-sulfate, nitrate, ammonium, and the sum of the other nss-ions), the water-soluble and insoluble organic matter (namely WSOM and WIOM, calculated as described in Section 2.2.1) plus the eBC mass (obtained by averaging on-line PSAP measurements). On a yearly average, the atmospheric concentration of the PM1 aerosol was quite low  $(0.89 \pm 0.56 \,\mu\text{g m}^{-3})$  average  $\pm$  standard deviation, n=87) but showing a noticeable variability across the year (min=0.06  $\mu$ g m<sup>-3</sup> for the sample 29/06/2019, max=3.00  $\mu$ g m<sup>-3</sup> for the sample 22/02/2020). On yearly average, the PM1 was mainly constituted in similar proportions of nss-sulfate (representing 33  $\pm$  13% of the total), seasalt (29  $\pm$  13%) and OM (28  $\pm$  16%, of which 22  $\pm$  14% represented by WSOM), the rest being accounted for by much smaller contributions of ammonium (4  $\pm$  2%), nitrate (1  $\pm$  1%), eBC (2  $\pm$  2%) and other non-sea salt ions (i.e., nss-K, nss-Mg and nss-Ca, amounting to 3  $\pm$  4% in total). Nonetheless, concentrations and relative contributions of the main chemical constituents experienced large variations along the year, highlighting a marked seasonality for PM1 composition, as shown in Figure 2.

During summer and early fall, OM reached its maximum both in absolute concentrations and in relative contributions to PM1  $(0.35 \pm 0.25 \,\mu g \,m^3)$ , representing  $44 \pm 18\%$  on summer average, of which  $35 \pm 16\%$  was accounted by WSOM), while seasalt and nss-sulfate touched their minima  $(22 \pm 16\% \,and \,24 \pm 11\% \,of \,PM1$  as summer averages, respectively). Conversely, winter and spring were dominated by nss-sulfate  $(41 \pm 10\% \,and \,42 \pm 8\% \,on$  winter and spring averages, respectively) and seasalt  $(31 \pm 12\% \,and \,28 \pm 8\% \,on$  winter and spring averages, respectively) while OM showed its minimum at the end of January (min=0.06  $\mu g \,m^3$  for the sample 29/06/2019, representing 6% of total PM1).

The concentrations of eBC were generally very low at GVB but rised in late winter and early spring reaching a maximum of  $0.14~\mu g~m^{-3}$  at the end of February 2020 (see also AMAP 2021). eBC and nss-sulfate share a prevalent anthropogenic origin and their late-winter/springtime increase is attributable to the Arctic Haze phenomenon (Shaw, 1995), which consists of continental pollution transported over long distances (Eurasia) during the months of the year of enhanced atmospheric meridional circulation (Abbatt et al., 2019; Song et al., 2021). By contrast, the summer peak of organics must be influenced by sources active in the Arctic region, including the marine biogenic sources associated by the increased phytoplankton activity in ice-free waters in absence of light limitation, and already discussed by Becagli et al., (2019). Anyway, it is noticeable that WSOM represented a substantial fraction of the total submicron aerosol mass also during winter-time, suggesting a variety of different sources impacting the site which are the focus of further investigations reported in the next paragraphs.

325

#### 3.2 WSOA chemical characterization and seasonality

discuss later.

The complementary use of the AMS and NMR techniques allows a comprehensive chemical description of the water-soluble organic aerosol (WSOA). The comparison between the AMS/H-NMR reconstructed WSOC and the concentrations of WSOC derived from the TOC analyzer shows that the both AMS and H-NMR analyses of WSOA were quantitative (slope of the regression line = 0.99 and 1.01, for AMS and H-NMR respectively; Figure S2).

The AMS offers quantitative chemical fingerprints of WSOA, including elemental ratios, while H-NMR provides detailed insights into functional groups in WSOA and identifies molecular markers. WSOA was highly oxygenated overall, with elemental ratios showing high yearly mean values (O:C of  $0.94 \pm 0.12$  and OM:OC of  $2.39 \pm 0.16$ ). Furthermore, the bulk elemental composition of WSOA remained relatively constant throughout the year, except during early summer, when a reduced O:C ratio and an increased H:C ratio were observed (Figure S5). Nevertheless, the composition of WSOA varied substantially over the year, as indicated by NMR analyses. Both the concentrations of molecular markers and the distribution of functional groups in WSOA exhibited substantial seasonal changes (Figures 3 and S4), MSA and alkyl-amines exhibited maximum concentrations during late spring and summer months and very small concentrations during the rest of the year (Figure S4). The season of high amines and MSA, approximately between April and August/September, corresponds to the season of high biological activity in the Fram strait and Norwegian and Barents seas. By contrast, HMS concentrations were more evenly distributed across the year, even though with a relative increase during spring. HMS, formed by the aqueousphase reactions between formaldehyde and the sulfur dioxide, is considered a tracer of cloud/aqueous-phase processing of anthropogenic emissions (Moch et al., 2020; Liu et al., 2021), which seems to be a relevant OA source/mechanism of formation during the whole year at the study site (Figure S4). Finally, levoglucosan showed variable concentrations, with a maximum in February possibly associated to residential heating emissions from the Eurasia and long-range transport in the Arctic haze (Gramlich et al., 2024), very small concentrations during the rest of the year but with sporadic peak concentrations in the

The H-NMR functional groups distribution also had variable contributions among the different seasons. Specifically, colder seasons were enriched in alcoxy (H-C-O) groups (39-42% in winter and fall respectively, compared to 24-28% during summer and spring). These H-NMR features have been previously associated to the abundance of primary emitted components such as sugars and polyols (e.g., glucose, sucrose, glycerol, etc.) (Facchini et al., 2008b; Liu et al., 2018; Decesari et al., 2020) and anhydrosugars, suggesting a higher influence of primary emission processes (with an unclear split between natural and anthropogenic sources) during the cold months. Between winter and the beginning of spring, during the Arctic haze season, the aromatic groups – although remaining a small component of the overall composition – reached a maximum, indicating an anthropogenic input. Conversely, the warm season was characterized by increasing contributions of unsaturated/branched aliphatic chains (H-C-C=), MSA and alkyl-amines, which are considered mostly secondary (Facchini et al., 2008a; Dall'Osto et al., 2019) indicating an increasing contribution of photochemical sources of oxidized organic aerosols.

summer (Figure S4). Such summer peaks can be associated to boreal forest wildfires (Bhattarai et al., 2019), as we will further

#### 3.3 WSOA Source Apportionment

The large inter-samples (intra-annual) chemical variability of WSOA in Ny-Ålesund based on the AMS and H-NMR analyses allowed the discrimination of different factors that were interpreted as WSOA sources (4 for AMS, 5 for NMR, for details see Supplementary Section S2, Figure S6-S18). Figure 4 reports the chemical profiles and contributions of these WSOA sources, showing remarkable agreement between the techniques. The specific AMS and NMR WSOA sources were associated between each other based on the correlations of their contribution time series (also reported in Table S3). Identification of the individual WSOA sources was further supported by the correlation of their contributions with the time series of molecular tracers, as reported in Table S4 and Figure S14. We also provide in Supplementary Table S3 specific AMS mass fragments identified in our dataset as characteristic of specific factors (i.e., meaning that more than 60% of their total measured mass is explained by a specific source-factor), which were also identified in previous studies.

Marine biogenic OA (Factor 1 – F1) identified by AMS was moderately oxygenated (OM:OC=2.02, O:C=0.63, H:C=1.76) and was characterized by sulfur-containing fragments (CH<sub>x</sub>SO<sub>y</sub>) from methane sulphonic acid (MSA) fragmentation (Zom et al., 2008; Chen et al., 2019). Specifically, marine OA explained a dominant fraction of the measured mass associated with many sulfur-containing ions such as CHS (44.98), CH<sub>2</sub>SO (61.98), CH<sub>3</sub>SO (62.99), CH<sub>2</sub>SO<sub>2</sub> (77.98), CH<sub>3</sub>SO<sub>2</sub> (78.98) and CH<sub>4</sub>SO<sub>3</sub> (95.99). Other important fragment ions in the profile have been related to sea spray emissions during phytoplankton blooms (CH<sub>3</sub>) and emissions from phytoplankton as well as kelp under heat stress (CH<sub>2</sub>O and CH<sub>3</sub>O) (Van Alstyne and Houser, 2003; Decesari et al., 2011; Faiola et al. 2015; Meador et al., 2017; Aguilera et al., 2022; Koteska et al., 2022; Saha and Fink, 2022). Consistently, marine OA reached maximum concentrations during summer with peak marine biological productivity and correlated very well with the concentrations of MSA and methyl-amines determined by ion chromatography (Pearson Correlation Coefficient R = 0.84 and 0.64, respectively) supporting the marine biogenic origin.

The marine biogenic OA was further separated based on H-NMR data into a SOA and POA contribution (Factor 1a and 1b – F1a and F1b, respectively), the sum of which correlates/compares very well with the AMS marine OA (Figure 4 and Table S3). Marine biogenic SOA (F1a) was dominated by MSA, with its specific singlet at 2.80 ppm of NMR chemical shift, and by low molecular weight methylamines, especially DMA, characterized by a singlet at 2.71 ppm. The predominance of these compounds and the correlation with the ion chromatography MSA (R = 0.98) supports a clear identification of marine biogenic secondary origin. Marine biogenic POA was characterized by a pattern of bands at 0.9, 1.3, 1.6, 1.8, 2.4 and 2.6 ppm of NMR chemical shift, corresponding to aliphatic methylenic chains with terminal methyl moieties and bound to a carbonyl/carboxyl group. Such chemical compounds share a linear aliphatic structure, with varying degrees of functionalization, and are attributable to degradation products of lipids including low-molecular weight fatty acids (LMW-FAs) and mixtures of other alkanoic acids (e.g., sebacate, suberate, adipate, caprylate, etc.). These features are typical of primarily emitted submicron seaspray particles, as shown via bubble bursting experiments of biologically-productive North Atlantic Ocean and Southern Ocean

sea-waters (Facchini et al., 2008b; Decesari et al., 2020; Paglione et al., 2024). However, with respect to previous studies identifying marine POA in ambient aerosols, the POA factor identified at Ny-Ålesund contained less methylenic long chains (in particular band at 1.3 ppm) and a higher degree of functionalization, pointing to a greater degree of oxidation/fragmentation of the low-molecular weight fatty acids (Figure S13). This was further reinforced by the good correlation between the contribution of marine biogenic POA and the concentrations of C3-C7 saturated dicarboxylic acids (Figure S14). Nevertheless, 435 the primary nature of this component was supported by the detection of saccharides (e.g. sucrose, glucose, and possibly ribose) and C3-C6 polyols (e.g. glycerol, D-threitol, grabitol, galactitol) (Figure S11 and S12). In addition, marine biogenic POA correlated well with sugars (i.e., glucose, sucrose, xylose, ribose) from HPLC-MS analyses (Figure S14). The impact of marine biogenic POA was also reflected in the AMS spectral fingerprint of marine OA with contributions of CxHyO1 (i.e., CHO at m/z 29.003) and CxHyO>1 fragments (e.g., C2H4O2 at m/z 60.021, usually associated to levoglucosan, but actually common to many other possible sugars (High-Resolution AMS Spectral Database. URL: http://cires.colorado.edu/jimenezgroup/HRAMSsd/, last access: 17 Jan 2025, Ulbrich et al., 2009; Bozzetti et al., 2016; Hu et al., 2018). Marine biogenic POA contributed most during summer, similar to the marine biogenic SOA, but its peak concentrations were shifted toward late summer. This timing likely coincided with the annual sea-ice minimum and the decaying phase of the algal bloom, a period previously linked to enhanced emissions of sea-spray organics (Rinaldi et al., 2013, O'Dowd et al, 2015).

origin. Similar CWT maps were found for marine OA and MSA in previous studies (Moschos et al., 2022; Pernov et al., 2024).

H-NMR marine SOA and POA have slightly different source areas: marine SOA exhibit an origin over a broader area but with a hotspot in the North Atlantic region south of 65°N and close to the British Islands: an area already widely studied for its potential of marine SOA formation (O'Dowd et al., 2004). The southern sources in ice-free waters experiencing spring algal blooms explain why marine SOA concentrations started to rise in Ny-Ålesund already in April and found first peaks in May. The source fingerprint of marine POA, instead, is well confined in the Arctic basin with hotspots in the Greenland and Barents Sea, the Fram Strait and Baffin Bay: such sources are therefore linked to sea-ice extension as well as to sea ice state (e.g., presence of leads, etc.), which explains the delayed formation of marine POA with respect to MSA. Arctic sea ice reached its annual summer minimum on September 18, 2019, according to NASA and the National Snow and Ice Data Center (NSIDC), but already in middle July it was substantially depleted in many coastal areas of Svalbard, Greenland and North-America (see Figure S15). Figure S15 (panel c) shows the ground type over which the backtrajectories of each PM1 sample were passing. In particular it highlights the higher fractional influence of sea-water vs sea-ice cover on the contributions of F1b (marine POA) as apportioned by NMR analysis, further supporting its interpretation.

The CWT maps for marine OA, as well as for its sub-fractions marine biogenic POA and SOA (Figure 5), supports their marine

Aged wildfires OA (Factor 2 – F2) was also moderately oxygenated (OM:OC=1.96, O:C=0.62, H:C=1.44), and characterized by both hydrocarbon (C<sub>x</sub>H<sub>y</sub>) and oxygenated (C<sub>x</sub>H<sub>y</sub>O<sub>1</sub> and C<sub>x</sub>H<sub>y</sub>O<sub>>1</sub>) fragment ions (see also supplementary Figure S7). In particular, a vast majority of fragments that were almost entirely apportioned to Factor 2 belong to the C<sub>x</sub>H<sub>y</sub>O<sub>>1</sub> family (e.g., CH<sub>5</sub>O<sub>2</sub>, C<sub>3</sub>H<sub>7</sub>O<sub>2</sub>, C<sub>3</sub>H<sub>3</sub>O<sub>3</sub>, C<sub>5</sub>H<sub>5</sub>O<sub>2</sub> and C<sub>5</sub>H<sub>6</sub>O<sub>2</sub>, at m/z 49.03, 75.04, 87.01, 97.03 and 98.04 respectively). H-NMR shows that

this component was characterized by branched/cyclic and poly-substituted aliphatic compounds, similar to atmospheric HULIS

(Humic-Like Substances) already associated with aged biomass burning emissions observed in many continental areas
(Decesari et al., 2000, Paglione et al., 2014a and 2014b, Decesari et al., 2014, Figure S16). Photochemical ageing is known to
progressively degrade levoglucosan and other polyols/anhydrosugars and phenolic compounds, leaving behind more refractory
aliphatic and aromatic compounds carrying carboxylic groups (Paglione et al., 2014a; Yazdani et al., 2023). F2 only showed
considerable concentrations during a few episodes in summer 2019 and later in spring 2020. Based on CWT analyses, we
identified plausible source areas of this OA component in eastern Europe but also in northern Siberia: an area which was
impacted by multiple huge wildfires during summer 2019 (Figure S17). So, Factor 2 mainly represents the impact of
oxidized/aged biomass burning particles from wildfires transported from boreal forests in Eurasia. Measured full-resolution
NMR spectra of samples representative of this factor (i.e., 04-Jul and 08-Jul), also showed signals of biogenic SOA formed
from the oxidation of terpenes and isoprene, including compounds like terebic acid, MBTCA (Methyl-butanetricarboxylic
Acid) and methyl-tetrols (Finessi et al., 2012; Zanca et al., 2017) (Figure S18). This suggests that the ageing of biomass burning
emissions was accompanied by terpene-derived SOA, potentially from direct emissions from the fire front or by
formation/accumulation of SOA during long-range transport over forested areas.

Arctic Haze OA (Factor 3 – F3) was highly oxygenated (OM:OC=2.26, O:C=0.83, H:C=) but also showed features in the AMS spectral profiles related to anthropogenic emissions such as from fossil fuel (i.e., hydrocarbon-like fragments C<sub>x</sub>H<sub>2x+1</sub> and C<sub>x</sub>H<sub>2x-1</sub> like C<sub>3</sub>H<sub>5</sub> & C<sub>3</sub>H<sub>7</sub>, C<sub>4</sub>H<sub>7</sub> & C<sub>4</sub>H<sub>9</sub>, C<sub>5</sub>H<sub>9</sub> & C<sub>5</sub>H<sub>11</sub>, at m/z 41 & 43, 55 & 57, 69 & 71 respectively, etc.) and biomass combustion (i.e. anhydrosugars fragments such as C<sub>2</sub>H<sub>4</sub>O<sub>2</sub> and C<sub>3</sub>H<sub>5</sub>O<sub>2</sub> at m/z 60.02 and 73.03, respectively). This was supported by the H-NMR analyses, showing that this factor was characterized by aromatics, levoglucosan and polysubstitued aliphatics but also by the hydroxy-methane sulfonate (HMS), a well-recognized significant anthropogenic contributor to midlatitude wintertime pollution (Moch et al., 2020; Liu et al., 2021), with its specific singlet at 4.39 ppm of NMR chemical shift. Factor 3 showed an increasing contribution during late winter and early spring at the peak of the Arctic haze season. Thus, this OA component also correlated quite well with eBC and nss-SO4 (Table S4), and has quite low concentrations until late winter like other phenolic compounds (i.e., vanillic acid, Figure S14). All this suggested that Arctic haze OA was related to long-range transport from anthropogenic emissions in the mid-latitude, including fossil fuel and biomass combustion. CWT analysis showed for Arctic haze OA a clear continental origin in Eurasia and, to a lesser extent, in North America, which represents the typical source fingerprint of Arctic haze at Svalbard. It is worth noticing that Arctic haze OA accounted for the largest contribution to winter-spring time concentrations of levoglucosan in Ny<sub>3</sub>Ålesund whereas in summer Aged wildfires OA was the major contributor (as shown also in Figure S17).

Background mix OA (Factor 4 – F4) in the AMS was dominated only by CO<sub>2</sub> and related fragment ions, making it extremely oxygenated (OM:OC = 2.88, O:C = 1.32, H:C = 1.13). Consequently, the source of this OA component remains uncertain based on its AMS chemical fingerprint. Factor 4 correlated strongly with oxalic acid and C<sub>1</sub>–C<sub>2</sub> monocarboxylic acids,

Fliminato:

indicating an enrichment in highly oxidized organic species consistent with its high oxidation state (Figure S7). The corresponding NMR factor revealed a mixture of anthropogenic signals (e.g., aromatics, HMS, traces of levoglucosan) and biogenic signals (e.g., aliphatics, polyols, amines), however with unclear attribution. Its seasonality was less pronounced than other OA types, though concentrations were lower during the dark months. The high oxidation state suggests extensive aging and distributed sources across vast geographical regions. The time trend similarities with maleic acid indicates a contribution from the aging of anthropogenic emissions, as maleic acid forms during the oxidation of aromatic volatile organic compounds (VOCs), Likewise possible signals of terpene-derived biogenic SOA from forest VOCs emissions can not be excluded in the NMR spectral profile of F4, even if not univocally identified. Simultaneously, overlaps with aminoacids time trends suggest mixing with a primary biogenic component (Figure S14). Background mix OA likely encompassed emissions from diverse sources fractions that the current chemical database using AMS and H-NMR cannot fully resolve. CWT maps for mixed background OA highlighted influences from extensive regions across the northern hemisphere, predominantly terrestrial but also including maritime zones in the North Atlantic. This broad, dispersed geographical impact, combined with spectral profiles and temporal trends, complicates definitive source attribution.

## 515 4 Sources contribution to OC

Since OC was to 71±18% water-soluble, only a relatively small fraction was missing in the source apportionment of WSOC. The missing 29% was quantified via multilinear regression (MLR) as already proposed in previous studies (Casotto et al., 2023; Cui et al., 2024) and described in Supplementary Section S3. Briefly, total OC mass was fit using a linear combination of the WSOC components (using AMS and NMR PMF-factors independently). The multiplicative coefficients attributed by the regression to each factor are considered to be recovery coefficients (RC), which are inversely related to solubility: higher coefficients mean that corresponding factor was less water soluble, and is associated with a higher fraction of insoluble OC.

The resulting coefficients are reported in Table S6.

For both the techniques, Factor 4 (the background mix OA) resulted to be a rather insoluble component. This is quite surprising, given that Factor 4 was also the most oxidized component (according to AMS elemental ratios), which is usually considered a hint of good water affinity. As a matter of fact, however, relatively large amounts of water-insoluble organic carbon can be found in the aerosols even at the most remote locations (e.g., Miyazaki et al. 2011).

The marine OA by AMS fitting seemed also quite insoluble. However, the fitting based on the NMR factors highlights how the most insoluble part of the marine components was related to the POA fraction, enriched (as already mentioned) of molecules characterized by limited solubility, such as lipids and possibly aminoacids and phenolic compounds. This agrees with our knowledge of marine POA production and composition according to which a large fraction is in fact water-insoluble (Facchini et al 2008b).

The Aged wildfires OA (characterized by HULIS features) and the Arctic haze OA components instead showed different coefficients for the fitting based on AMS and NMR factors: slightly higher than 1 for AMS (i.e., 1.21 and 1.56, respectively), while the highest values for NMR (i.e., 2.22 and 2.04, respectively). This might result from the presence in these components of molecules characterized by an intermediate solubility and by the abundance of functional groups not bearing detectable protons (such as carbonyls, fully substituted aryls and also carboxylic acids) which are under-resolved by NMR. These features are also consistent with the presence in Arctic haze OA of anthropogenic products coming from fossil fuel combustions (i.e., hydrocarbon-like fragments in AMS profile and aromatics, alkyls, alkenes, polysubstituted aliphatics in NMR profile).

Given the good agreement between the AMS and NMR reconstructions of the total marine WSOC fraction (see Figure 4), we further split marine OA by AMS based on the relative NMR contributions of its primary and secondary marine components (as further described in Supplementary Section S3). Figure 6 reports the scaled contributions of the NMR and AMS factors to

total OC at Ny-Ålesund along the whole year showing again a marked seasonality, while Table 1 details their yearly and seasonal averages and variability (i.e., standard deviations). Looking at the relative contributions, the marine biogenic components leaded the submicrometric OA mass during summer (31% and 44% as the sum between marine SOA and POA in AMS and NMR, respectively) and spring (10% and 17%), as a consequence of the increased biological activity characterizing the warmest seasons. During winter the marine SOA component disappeared almost completely (0.3% and 0.7% in AMS and NMR, respectively), following the seasonal reduction of solar light and photo-oxidative conditions, while marine POA factor had still little but not negligible relative contributions (1% and 3%). This finding supports the presence of natural primary sources still operating at the site during winter, as already suggested by the functional groups distribution given above (Figure

sources still operating at the site during winter, as already suggested by the functional groups distribution given above (Figure 3). However, it remains unclear to what extent marine POA continued to form over ice-covered oceanic regions. It is possible that the marine POA factor in cold months actually traces other organic components rich of polyols such as fungal spores, whose chemical tracers actually peak in the fall season (Figure S14).

A relevant contribution during summer (28% and 27% in AMS and NMR, respectively) was also represented by the aged aerosols (for NMR enriched of HULIS compounds of continental origin) attributed to long-range transport of biomass burning and biogenic SOA particles probably from wildfires of Eurasian boreal forests.

Arctic haze contribution, characterized by aged anthropogenic compounds linked with fossil fuel and biomass combustion mainly coming from continental polluted Eurasian mid-latitudes, was identified and quantified contributing to relevant portion of OA during winter/spring (53% and 70% in winter, 31% and 27% in spring for AMS and NMR, respectively) becoming significantly less important in the summer months (7% and 6%). Factor 4, background mix OA, had lower time variability during the year than other components (40 and 28% in AMS and NMR, respectively), even though it showed a reduction in the concentration during winter time (35% and 19%).

Moschos et al. (2022) attributed a distinct, albeit minor, contribution of BSOA (biogenic secondary organic aerosol) to organic carbon (OC) during the polar day at Ny-Ålesund - 9% at Gruvebadet and 19% at Zeppelin. In contrast to their findings, our PMF analysis was unable to clearly isolate and quantify a specific BSOA contribution within our dataset. This is likely due to strong co-variation between BSOA and other particles that are co-emitted, formed, and transported to Ny-Ålesund from

Fliminato

common source regions, primarily the boreal forests of Eurasia. Nevertheless, the overall picture emerging from our study remains broadly consistent with the 9–19% BSOA contribution reported by Moschos et al. (2022). These BSOA components are likely incorporated year-round into varying portions of our F2 (aged wildfire OA) and/or F4 (background mix) factors, which represent up to 28% and 34% of summertime organic PM<sub>1</sub>, respectively,

Although the AMS and NMR showed an overall good agreement in OC source apportionment, some discrepancies could be noticed in the relative contributions of specific components to the aerosol OC (Table 1). While the total marine and the wildfires OA fractions agreed quite well, a greater contribution for Factor 4 (background OA) with respect to Factor 3 (Arctic haze OA)

OA fractions agreed quite well, a greater contribution for Factor 4 (background OA) with respect to Factor 3 (Arctic haze OA) was derived by AMS when comparing to NMR (for background OA 40 versus 28% and for Arctic haze OA 28 versus 33% on yearly average, for AMS and NMR respectively). We believe that such discrepancies, likely related to the different sensitivity of the two instruments to specific organic mixtures (see Sect. 2.2.1 and 2.2.2), provide a measure of the level of bias one may encounter when relying on a single technique to characterize OA, representing a further relevant output of this study. And despite these discrepancies, the overall agreement between NMR and AMS characterizations highlights the robustness of the study's findings and reveals a consistent picture of the main organic submicron aerosol sources in Ny-Ålesund and their

5 Conclusions

seasonality.

A complete yearly set of PM1 samples from Ny-Ålesund (May 2019-June 2020) was analyzed for the first time in parallel by H-NMR spectroscopy and off-line HR-ToF-AMS, alongside bulk chemical analyses. About 90% of PM1 mass was apportioned into organic mass, nss-sulfate and seasalt in similar proportions: 28%, 33% and 29%, on a yearly basis. Noticeably, organic aerosol was the major PM1 component during summer, accounting for 44% of the PM1 mass.

The applied organic aerosol characterization techniques provided a picture of the evolution of bulk organic aerosol features through the year. From the AMS perspective, summertime OA was Jess oxidized, probably reflecting the influence of a higher contribution from local (Arctic) sources, while it tended to be generally more oxidized during fall, with spring and winter representing intermediate conditions, under the variable effect of long-distance terrestrial sources including anthropogenic pollution at the mid-latitudes and boreal forests.

Organic aerosol source apportionment applied in parallel showed exceptionally good agreement between the two techniques which demonstrates the good complementary between H-NMR and AMS, especially when the latter is run off-line. As a corollary, when removing the uncertainties associated with the sampling and the analyses are performed on the same filters, even very different spectroscopic techniques like H-NMR and AMS can lead to substantially consistent results in organic source apportionment, supporting the idea that "factors" approximate real chemical classes. For the same reason, the different selectivity of one technique with respect to given chemical structures can be exploited to achieve a better understanding of the chemical nature of "factors" with respect to when employing a single analytical method. This is especially useful for AMS

Fliminato

that, even if well established in the atmospheric chemistry community, can be limited in providing source attribution for the oxidized organic aerosol (OOA) types, and would therefore gain high benefit from the integration with other methodologies. In this study, in particular, NMR supported the interpretation of the AMS factor profiles (which showed a lower variability in spectral features than NMR ones) and allowed the separation of the marine source in two different contributions, primary and secondary, enhancing our source apportionment capability in this study. Nevertheless, our results also show the limits of OA source apportionment in remote locations, where aerosols are often transported from long distances, underwent heavy atmospheric chemical processing (ageing) and are well mixed. Indeed, for one of the factors (F4 –background OA), accounting for an important 28-40% of OA, it was not possible to attribute a specific source or formation process, nor even to classify it as anthropogenic or natural and it could comprise several distinct, unresolved contributions.

Both anthropogenic and biogenic sources were shown to contribute to the sub-micrometer OA load, with a marked seasonality:

- Biogenic aerosols (both primary, 1-30% and secondary, 1-14%, as min-max annual relative contributions to OA respectively), dominating during summer (31-44% in total) were associated to sulfur compounds, amines and fatty acids with their degradation products (DCAs) from the marine environment.
- Aged anthropogenic aerosol in summer was attributed to long-range transport of biomass burning particles from Eurasian forests (7-28%, annual min-max respectively); for NMR this component looked like to be enriched of HULIS compounds of continental origin.

Arctic haze contribution (6-70%, annual min-max respectively), characterized by aged anthropogenic compounds
linked with fossil fuel and biomass combustion, was identified and quantified contributing to relevant portion of OA
during late-winter/early-spring (27-70%, winter-spring time min-max respectively).

These findings are consistent with the seasonality of anthropogenic and biogenic OA sources described by Moschos et al. (2022) in a previous pan-Arctic study and confirm the summertime relevance of biogenic sources also for the sub-micrometer OA fraction. The present results also evidence that the European Arctic is heavily impacted by anthropogenic sub-micrometer OAs, in particular from fall to spring. This shows how atmospheric emissions from lower latitudes are key drivers of Arctic air quality and, ultimately, climate. With respect to the Moschos et al. (2022) study, we could not separate a distinct fraction for primary biological particles (PBOA), probably because of their small contribution in PM1. At the same time, the employ of H-NMR factor analysis enabled to detail the chemical nature of the marine OA showing a new component associated with the atmospheric life cycle of marine POA beside the already known contribution from MSA. Marine OAs were shown to dominate organic composition in summertime outside outbreaks of wildfire plumes, when concentrations peaked between 0.1 and 0.2 µgC m<sup>-3</sup> in PM1 which in turn compare with the concentration levels reached at the end of the winter in full Arctic haze season: these findings once again highlight the importance of the research on the climate-ecological feedbacks on the production of aerosols and aerosol precursors from the marine biota in Arctic and sub-Arctic environments. As long as the transport of anthropogenic aerosols rich of organic carbon, black carbon and nss-sulfate to the Arctic will continue to decrease, the importance of climate feedbacks on the biogenic sources as well as on fire activity will increasingly shape the landscape of aerosol-climate interactions in the coming decades.

#### Data availability

Data are available at IADC public repository (https://doi.org/10.71761/0e110925-1f3d-4013-b048-e5a47ca3be6f) eBC data were obtained from Mazzola and Gilardoni (2022).

## Competing interests

The authors declare that they have no conflict of interest.

#### Author contributions

M.P and M.Rin. designed the research; M.P., M.Maz., and M.Rin. organized the field campaign; M.P., and M.Maz. collected the aerosol samples; M.P., Y.F., M.Rus., M.I.M., E.B. and D.F. performed the chemical analyses; M.P. and M.Rin. performed the factor analysis; S.D., A.M., E.T, K.D. and F.C. contributed to the factor analysis discussion and correction; K.M elaborated back-trajectories and maps. M.P., K.D., M.R., and S.D. wrote the paper. All the authors contributed the scientific discussion and paper revision.

# Acknowledgments

Financial support was provided by the European Commission: H2020 Research and innovation program, project FORCeS (grant no. 821205) and by the Italian Joint Research Center ENI-CNR "Aldo Pontremoli" within the ENI-CNR Joint Research Agreement. Support from CleanCloud (Horizon Europe, grant no. 101137639) is also greatly acknowledged. KRD acknowledges support by SNS PZPGP2\_201992 grant. This research was also supported by the Italian Ministry of University and Research (MIUR) within the framework of the Arctic Research Program of Italy of the project "BETHA-NyÅ" - 670 Boundary layer evolution through harmonization of aerosol measurements at Ny-Ålesund research station (PRA2021-0020). The authors thank CNR and the staff of the Arctic Station Dirigibile Italia for their support: F. Di Bona, F. Bruschi, P. Gravina, R. Nardin, M. Casula, M. Pallottini. We thank also A. Cavaliere and G. Verazzo from CNR-ISP (Bologna) for helpful

database.

discussions about data fairness and for uploading and organizing the study dataset into the Italian Arctic Data Center (IADC)

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

Figure 1. Maps of the study area.

Figure 2. PM1 loadings and chemical composition at Ny-Ålesund during the whole study period. Panels (a) and (b) show the respective mass concentrations and relative contribution of the different chemical species measured in each sample. Pie charts in panel (c) and (d) report the average relative contributions for the whole year and for the different seasons of the campaign.

Figure 3. Water-soluble OC concentrations and composition in term of H-NMR functional groups. Panel (a) and (b) show respectively the mass concentrations and the relative contributions of the different functional groups identified and quantified by H-NMR in each sample (expressed in µgC m-3). Pie charts in panel (c) and (d) show the average relative contributions for the whole year and for the different seasons of the campaign.

Figure 4. HR-AMS (left-side) and H-NMR (right-side) factors spectral profiles and contributions (center) resulting by PMF analysis of the Ny-Alesund 2019-20 dataset. On the left-side, AMS HR mass spectral profiles are shown as normalized fragment intensities (with average atomic ratios in the boxes), where the fragments are color-coded with the families (in the legend); some specific fragments are highlighted as representative of individual or groups of tracer compounds (such as CH<sub>2</sub>SO<sub>2</sub> and CH<sub>2</sub>SO<sub>2</sub> for MSA: methane-sulfonate, or C<sub>2</sub>H<sub>2</sub>O<sub>2</sub> for levoglucosan and anydrosugars, C<sub>n</sub>H<sub>n+1</sub> and C<sub>n</sub>H<sub>n-1</sub> for hydrocarbons). In the center, contributions time series of NMR Fla and Flb are stacked one on top of the other. On the right-side, H-NMR peaks of individual compounds (MSA: methane-sulfonate; HMS: hydroxymethane sulfonate; Levoglucosan) are specified in the profiles, along with the band of functionalities and unresolved mixtures: LMW-FAs (low-molecular weight fatty acids), sugars and polyols indicating saccharides such as sucrose, glucose, and possibly ribose and C3-C6 polyols (e.g. glycerol, D-threitol, arabitol, galactitol). Pie charts report the H-NMR functional groups distribution of each factor.

Figure 5. CWT maps of the OA components identified by the factor analysis of HR-AMS (left side) and H-NMR (right side) spectra at the Ny-Alesund sampling site (blue square in the maps). The colors show which air masses along the back trajectories have, on average, higher concentrations (expressed in µgC m-3) evidenced by darker colors. Red dots represent values at or above the 90th percentile, the most probable regions associated with high concentrations at the sampling site.

Figure 6. Contributions of the H-NMR (left-side) and AMS (right-side) factors identified by PMF to the total Organic Carbon (OC) as resulting from Multilinear regression analysis. Histograms show the mass concentrations (µgC m-3, panels a. and e.) and the relative contributions (%, panels b. and f.) of each factor in each sample. Pie charts report the average relative contributions for the whole year (panels c. and g.) and the different seasons (panels d. and h.).

Table 1. annual and seasonal ranges of the average mass contributions to total OA of the sources identified by AMS and H-NMR source apportionment. Average values (in black) are followed by standard deviations (in grey).

| -            | marine SOA     |                  | marine POA     |                | Aged wildfires OA |                | Artic haze OA  |                | Background OA  |                |
|--------------|----------------|------------------|----------------|----------------|-------------------|----------------|----------------|----------------|----------------|----------------|
| Ave ±std.dev | AMS            | NMR              | AMS            | NMR            | AMS               | NMR            | AMS            | NMR            | AMS            | NMR            |
| SUMMER       | <b>11</b> ±11% | <b>14</b> ±11%   | <b>20</b> ±15% | <b>30</b> ±20% | 28 ±21%           | <b>27</b> ±22% | 7 ±8%          | 6 ±6%          | <b>34</b> ±25% | <b>23</b> ±18% |
| FALL         | 1.9 ±3.1%      | <b>2.1</b> ±1.7% | <b>1</b> ±2%   | <b>7</b> ±8%   | <b>25</b> ±10%    | <b>17</b> ±16% | <b>24</b> ±24% | <b>31</b> ±15% | 48 ±21%        | <b>43</b> ±19% |
| WINTER       | 0.3 ±0.8%      | 0.7 ±0.5%        | 1 ±3.3%        | 3 ±3%          | <b>11</b> ±6%     | <b>7</b> ±8%   | 53 ±21%        | <b>70</b> ±9%  | <b>35</b> ±20% | <b>19</b> ±12% |
| SPRING       | 9 ±15%         | <b>10</b> ±13%   | 2 ±3%          | <b>7</b> ±5%   | <b>14</b> ±9%     | <b>33</b> ±17% | <b>31</b> ±23% | <b>27</b> ±20% | <b>44</b> ±16% | <b>23</b> ±17% |
| Whole        | 5 ±10%         | 6 ±10%           | <b>7</b> ±12%  | <b>12</b> ±16% | <b>20</b> ±15%    | <b>21</b> ±19% | <b>28</b> ±26% | <b>33</b> ±27% | 40 ±22%        | <b>28</b> ±19% |

Y