# Peer review of "Unraveling Arctic submicron organic aerosol sources: a year-long study by H-NMR and AMS in Ny-Ålesund, Svalbard"

_EGUsphere, 2025_

## Author Comment (AC1)

We thank the Reviewer #1 for the helpful and constructive comments, which have led to significant improvements in the manuscript. We have carefully revised the text. Our point-by-point replies are given below (blue), following the referees' comments (black). Changes to the manuscript are marked with green.

**Reviewer 1**:

Paglione et al. present a detailed analysis of PM1 filter samples collected over the course of a year at Ny-Ålesund, Svalbard. Through offline analysis via NMR, AMS, and other techniques, the authors characterize the water soluble fraction of PM1. The AMS and NMR datasets independently yielded similar PMF factors related to marine OA (further resolved as POA and SOA for the NMR data), wildfire OA, Arctic haze OA, and general atmospheric background OA. Factor identities were supported by correlations with tracers, as well as source analysis by backward airmass trajectories. Identified factors align with current understanding of Arctic aerosol sources composition and sources, but with added chemical information and long-term measurements to further assess seasonality. This study by Paglione et al. makes a meaningful contribution to the field and understanding of Arctic aerosol composition. I recommend this manuscript for publication following revisions in response to my following comments.

Response: We appreciate the general Reviewer's positive feedback on the manuscript relevance.

General Comments:

The use of acronyms should be cleaned up throughout the manuscript. For example, WSOC is defined in the introduction (line 93), but why reiterate the meaning of WSOC in section 2.2 (line 129) rather than 2.1 (line 114)? Similarly, WSOA is indirectly defined in the abstract as "…PM1, with a specific focus on its water-soluble organic fraction (WSOA)…" (line 24), later indirectly again as "…organic aerosol (OA) by analyzing its water-soluble fraction (WSOA)…", and finally explicitly "…water-soluble organic aerosol (WSOA)" (line 318). I recommend being explicit with acronyms to avoid potential confusion. Long-range transport is defined twice back-to-back on lines 45 and 56. BSOA is defined on line 54, "biogenic secondary organic aerosol" is redefined as just "SOA" on line 74, then "biogenic secondary organic aerosol" is used with no acronym on line 78. This list is non-exhaustive. For clarity, be explicit and consistent with acronym definitions and use.

Response: we thank the Referee for his/her careful review of the text. We have revised the acronyms to be consistent and clearer along the whole paper accordingly.

Specific Comments:

Line 24 – for clarity, I recommend defining WSOA with "water soluble organic aerosol" fully written out, rather than relying on the reader to interpret "…PM1, with a specific focus on its water-soluble organic fraction…"

Response: since we prefer to avoid too many acronyms in the abstract we decided to remove the WSOA abbreviation here and to use and make it fully explicit later in the following text.

Line 29 – Can you clarify what the percentages refer to? I assume PM1 contributions by mass?

Response: actually, the percentages here are the relative contributions to total OA mass. To make the point clearer, we rephrased the previous sentence (Lines 27-28) as follows:

Positive Matrix Factorization (PMF) source apportionment identifies consistent OA sources from the H-NMR and AMS datasets, showing a pronounced seasonality in their relative contributions to total OA mass.

Line 38 – I suggest rephrasing "…given the fast rate of temperature growth in this area…" to specify that the faster temperature increase is relative to the rest of the globe.

Response: we thank the Referee for his/her suggestion. We rephrased the sentence as follows:

"…given the faster-than-global warming rate of this area"

Lines 47-48 – While long-range transport of anthropogenic pollution may be declining, it is also important to acknowledge the increasing local pollution from resource extraction and shipping (e.g., Peters et al., 2011; Pizzolato et al., 2016).

Response: we thank the Referee for his/her suggestion. We rephrased the sentence accordingly as follows:

"In turn, local pollution from resource extraction and shipping (e.g., Peters et al., 2011; Pizzolato et al., 2016) as well as, most importantly, natural aerosol sources may become increasingly significant in the future."

Line 64-65 – "Therefore, current efforts have so far been unable to provide an understanding of the sources and formation pathways of the pan-Arctic OAs in different seasons." This is a strong statement, somewhat misleading, and contradicted in the following paragraphs. More long-term measurements are certainly necessary, but this statement implies there is no knowledge on Arctic OA sources/formation/seasonality. This statement should be rephrased.

Response: we thank the Referee for his/her comment. The authors' intention was not to discredit or minimize the efforts made so far to understand the aerosol sources in the Arctic, but only to highlight the still existing gaps in the organic aerosol sources comprehensive chemical characterization (in term of speciation of their total mass) and seasonality. The list of works and achievements in the following sentences aims precisely to recognize the work done so far and to explain how our work fits into a wider literature on the subject and which novelties it could add. In any case, we agree that the highlighted statement appears too strong and misleading and so we decided to rephrase it as follows:

"Therefore, despite current efforts, still many gaps prevent a comprehensive understanding of the pan-Arctic OAs sources and formation pathways in different seasons."

Line 93 – Would WSOA fit better than WSOC since you get non-carbon species in the AMS and IC?

Response: we thank the Referee for his/her suggestion. Actually, combining all the techniques, in the end we estimate contributions on total OA mass and so we decided to rephrase the sentence as follows:

"Here, we present the first year-long investigation combining AMS, H-NMR, ion chromatography (IC), organic and elemental carbon (OC and EC, respectively) measurements to characterize OA in the high Arctic. Conducted at the Gruvebadet Laboratory in Ny-Ålesund, Svalbard, this study provides seasonal insights into anthropogenic and biogenic contributions to Arctic OA."

Section 2 - You provide LODs for the EC/OC method (line 124), but what about LODs for your other methods? Did you perform any replicate measurements (e.g., did you run multiple aliquots of the extracts through the IC for an average and standard deviation?)

Response: following the request of the Reviewer, LODs of each species are now reported in Table S1. For IC and TOC-analyzer data, the LODs are calculated (for each species) based on the average and standard deviation of the five field blanks analyzed. In particular LOD = BLK_mean + 2* BLK_std.dev. No systematic replicates where analyzed due to the limited amount of material available for all the analyses, but for the IC measurements some samples were routinely replicated with different dilution factors in the case some species had concentrations out of the calibration range.

Table S1. Limit of Detection (LOD) for each species measured by Ion Chromatography (IC) and for water-soluble organic carbon (WSOC) by TOC analyzer. The LODs are calculated based on the average and standard deviation of the five field blanks analyzed. In particular LOD = BLK_mean + 2* BLK_std.dev.

| | Na | NH4 | ma | K | dma | tma | Mg | Ca | ace | for | MSA | Cl | NO2 | NO3 | SO4 | oxa | WSOC |
|---|---|---|---|---|---|---|---|---|---|---|---|---|---|---|---|---|---|
| | \multicolumn{16}{c}{IC ($\mu$g/mL)} | TOC-analyzer ($\mu$g/mL) |
| LOD | 0.36 | 0.03 | 0.05 | 0.05 | 0.05 | 0.05 | 0.004 | 0.06 | 0.02 | 0.02 | 0.02 | 0.21 | 0.01 | 0.03 | 0.12 | 0.05 | 1.16 |

Line 104 – Does the sampler have a model (Echo?)?

Response: we added the information on the sampler model.

Line 105 – Please provide more details on washing and baking the filters.

Response: synthetic information added in the revised paragraph.

Line 108 – What was the range of collection times?

Response: please find here below the detailed statistics of the sampling times. We added in the main text the information about mean ± standard deviation.

|  | sampling time (h) |
|---|---|
| mean | 84 |
| st.dev | 9 |
|  |  |
| max | 96 |
| 90th percentile | 89 |
| 75th percentile | 87 |
| median | 86 |
| 25th percentile | 84 |
| 10th percentile | 81 |
| min | 19 |

Line 108 – Were the field blanks evenly spaced throughout the campaign? Could you mark them in Fig. S1?

Response: Field blanks were collected randomly through the whole year about every 1 month, but not all of them were analyzed. We added an indication of the collection time of field blanks used in this work in the revised version of Fig S1 (and here below). We acknowledge that the blanks analyzed do not cover the entire period homogeneously, but, given that the materials and sampling system/procedure have not changed over time, we also expect that there have been no substantial variations over the period.

[Figure]

Figure S1. Sampling periods of the PM1 filters collected at Gruvebadet, Ny-Ålesund , and subsequently analyzed by H-NMR (dark-light green bars) and HR-TOF-AMS (dark-light blue bars). Field blank filters collection time is indicated by grey lines and names.

Line 111-113 – Can you clarify the filter portions? I could interpret this as "half of each filter was used in Bologna, the other half at Zurich" or "half of each filter was used in Bologna, while the

other half was cut and a part of that other half was sent to Zurich." I assume the former interpretation is the intention.

Response: We thank the Referee for highlighting the confusion in this information. Actually, in Bologna was analyzed half of the sampled filters, while in Zurich was shipped and analyzed a quarter of them. We rephrased the sentence to make clearer the point, as follows:

"About half of each filter was used for the off-line characterization by TOC-analyzer, Ion-chromatography and H-NMR (as described below) performed in Bologna, while a quarter was shipped to PSI laboratories in Zurich, Switzerland for AMS measurements and carbonaceous content quantifications (both WSOC by TOC analyzer and EC/OC on filters by Sunset, as detailed below)."

Line 151-159 – The sentence describing the identified functional groups is difficult to read. This may be easier to understand as a table. If possible, the addition of approximate chemical shift ranges would be useful too since NMR is less common for atmospheric measurements.

Response: we thank the Referee for his/her suggestion. We added along the text information about specific chemical shifts representing the different functional groups and also a new Supplementary Table (Table S2) reporting a comprehensive list and description of the functional groups, molecular species and categories of compounds identified by H-NMR spectra analysis in this study (as reported also here below)

**Table S2**. H-NMR identified/measured functional groups/chemical species/categories. *Functional groups are in *italic*. **Categories including some of the other species specifically identified are in underlined italic

| name of the species/ functional group*/ category of compounds** | ID of the species/ functional group | chemical shifts used for identification & quantification | examples for molecules | possible origin/source | references |
|---|---|---|---|---|---|
| *aromatic protons* | Ar-H | band 6.5-8.5 ppm | phenols, nitro-phenols [...] | biomass burning, [...] | Decesari et al., 2001; Tagliavini 2006; Decesari et al., 2007; Chalbot and Kavouras, 2014 |
| *anomeric and/or vinyl protons* | O-CH-O | band 6-6.5 ppm | vinylic protons of not completely oxidized isoprene and terpenes derivatives, of products of aromatic-rings opening (e.g., maleic acid), or anomeric protons of sugars derivatives (glucose, sucrose, levoglucosan, glucuronic acid, etc.) | biogenic marine mostly primary | Decesari et al., 2001; Claeys et al. 2004; Schkolnik & Rudich, 2005; Tagliavini 2006; Decesari et al., 2007; Chalbot and Kavouras, 2014 |
| *hydroxyl/alkoxy groups* | H-C-O | band 3.2-4.5 ppm | aliphatic alcohols, polyhols, saccharides, ethers, and esters | biogenic marine primary | Chalbot and Kavouras, 2014 |
| *benzyls and acyls/ amines, sulfonates* | H-C-C= / H-C-X (X≠O) | band 1.8-3.2 ppm | protons bound to aliphatic carbon atoms adjacent to unsaturated groups like alkenes (allylic protons), carbonyl or imino groups (heteroallylic protons) or aromatic rings (benzylic protons) | biogenic/anthropogenic mostly secondary | Decesari et al., 2001; Graham et al., 2002; Decesari et al., 2007; Chalbot and Kavouras, 2014 |
| *unfunctionalized alkylic protons* | H-C | band 0.5-1.8 ppm | methyls (CH3), methylenes (CH2), and methynes (CH) groups of several possible molecules: fatty acids chains, alkylic portion of biogenic terpenes, etc. | biogenic/anthropogenic primary/secondary | Decesari et al., 2001; Graham et al., 2002; Decesari et al., 2007; Chalbot and Kavouras, 2014 |
| hydroxymethansulfopnic acid | HMSA | singlet at 4.39 ppm | | anthropogenic secondary | Suzuki et al., 2001; Gilardoni et al., 2016; Brege et al 2018 |
| methane-sufonate | MSA | singlet at 2.80 ppm | | biogenic marine secondary | Suzuki et al., 2001; Facchini et al., 2008a; Decesari et al., 2020 |
| di-methylamine | DMA | singlet at 2.72 ppm | | biogenic marine secondary | Suzuki et al., 2001; Facchini et al., 2008a |
| tri-methylamine | TMA | singlet at 2.89 ppm | | biogenic marine secondary | Suzuki et al., 2001; Facchini et al., 2008a |
| *anhydrosugars* | | anomeric singlet between 5.40-5.45 ppm & specific structures between 3.5 and 4.6 ppm | levoglucosan, mannosan, galactosan and anomeric-C anhydroderivatives from cellulose/lignin combustion | biomass burning | Tagliavini et al., 2006; Pietrogrande et al., 2017 |
| levoglucosan | levo | anomeric singlet at 5.45 ppm & specific structures between 3.5 and 4.6 ppm | | biomass burning | Tagliavini et al., 2006; Paglione et al., 2014a&b; Pietrogrande et al., 2017 |
| *saccharides* | Sac | used synonymously for compounds carrying H-C-O groups in unresolved mixtures but when also anomeric protons (O-CH-O) are present | glucose, sucrose and other sugars structurally similar not unequivocally identified | biogenic marine primary | Graham et al., 2002; Facchini et al., 2008b; Decesari et al., 2011; Decesari et al., 2020; Liu et al., 2018; Dall'osto et al., 2022°; Paglione et al., 2024 |
| glucose | Gls | anomeric doublet at 5.22 ppm & specific structures between 3.5 and 4.2 ppm (not quantified but possibly quantifiable @5.22 ppm) | | biogenic marine primary | Decesari et al., 2020; Dall'Osto et al., 2022b |
| sucrose | Suc | anomeric doublet at 5.40 ppm & specific structures between 3.5 and 4.2 ppm (not quantified but possibly quantifiable @5.40 ppm) | | biogenic marine primary | Decesari et al., 2020; Dall'Osto et al., 2022b |
| ribose | Rib | anomeric doublet at 5.37 and 5.24 ppm & specific structures between 3.6 and 4.2 ppm (not quantified) | | biogenic marine primary | Suggested by this study (to be confirmed) |
| *polyols* | | unresolved mixture not quantified (including glycerol and D-threitol) | glycerol, threitol, erytritol and structurally similar molecules not unequivocally identified | | |
| glycerol | Gly | specific structures at 3.55, 3.66 & 3.77 ppm (not quantified but possibly quantifiable @ 3.55 ppm) | | biogenic marine primary | Decesari et al., 2020; Dall'Osto et al., 2022b |
| D-threitol | D-th | specific structures between 3.6 - 3.7 ppm (not quantified) | | biogenic marine primary | suggested by Paglione et al., 2024 (to be confirmed) |
| arabitol | Arab | specific structures between 3.6 - 4 ppm (not quantified) | | biogenic marine primary | Suggested by this study (to be confirmed) |
| galacticol | Gal | specific structures between 3.7 - 4 ppm (not quantified) | | biogenic marine primary | Suggested by this study (to be confirmed) |
| *phenolic compounds* | PCs | unresolved resonances between 6.5 – 7.2 ppm | Phenol and other compounds consisting of one or more hydroxyl groups (−OH) bonded directly to an aromatic ring (e.g., vanillic acid, etc.) | biomass burning [...] | Decesari et al., 2007; Chalbot and Kavouras, 2014 |
| *low-molecular weight fatty acids or "lipids"* | LMW-FA | unresolved complex resonances at 0.9, 1.3, and 1.6 ppm in the H-C spectral region | fatty acids (free or bound) from degraded/oxidized lipids (e.g. caproate, caprylate, suberate, sebacate, etc.) and similar compounds owning a chemical structures of alkanoic acids. | biogenic marine primary | Graham et al., 2002; Facchini et al., 2008b; Decesari et al., 2011; Decesari et al, 2020; Liu et al., 2018 |
| *biogenic SOA* | BSOA | Series of singlets/doublets between 0.9 – 1.6 ppm | compounds formed from the oxidation of terpenes and isoprene, including terebic acid, MBTCA (Methyl-butanetricarboxylic Acid) and methyl-tetrols | biogenic terrestrial secondary | Finessi et al., 2012; Zanca et al., 2017 |

Line 163 – I suggest adding a reference to Fig. S2a for "on average 30%."

Response: done, thanks.

Section 2.4 – Which meteorology data did you use? Did you use isobaric or isentropic trajectories?

Response: We used the archived Global Data Assimilation System (GDAS1) for trajectory calculations. Regarding the type of vertical motion method, we chose the default "Model vertical velocity", which uses the vertical velocity field from meteorological data. We added the following clause to the revised manuscript:

"The trajectory calculations were driven by meteorological data from the archived Global Data Assimilation System (GDAS1; 1° × 1°) of the National Centers for Environmental Prediction (NCEP (ftp://arlftp.arlhq.noaa.gov/pub/archives/gdas1, last access: 01 August 2022)."

Section 2.4 – The authors should acknowledge the general uncertainties of backward airmass trajectories in the Arctic due to a lack of meteorological measurements to constrain the model (e.g., Harris et al., 2005; Kahl, 1993).

Response: to acknowledge this uncertainty we added the references suggested and a sentence at L262 of the revised text, as follows:

"Despite the general uncertainties of the backward air mass trajectories in the Arctic due to the lack of meteorological measurements to constrain the model (e.g., Harris et al., 2005; Kahl, 1993), this approach is widely used in supporting identification of the source area of the different aerosol components measured at a receptor site."

Section 2.5 – Perhaps I missed it in the SI, but what fraction of the AMS and NMR signals is accounted for using PMF? In other words, how much of the measured signal on each instrument is not included in any of the factors?

Response: this information can be retrieved from the so-called "residuals" of the PMF model. The "unexplained" mass/signal can be quantified subtracting the residuals to the initial input mass/signal. In the case of our datasets, the residuals were -2±5% and -8±18% of total input-signal (measured WSOC mass conc.) on average of the whole year for AMS and NMR datasets, respectively. The negative average value indicates a general (very small) overestimation of the models. The unexplained signal varies depending on the season and in particular higher residuals can be found in specific samples at very low and/or very high concentrations (also because a higher uncertainty is attributed to them in the input error-matrix), so especially during fall and winter. For the Referee, we report here below the time series of the quantitative residuals together with the PMF-factors for both AMS and NMR (Figure R1.1). But we do not add any information to the manuscript because in PMF standard applications, considering the intrinsic uncertainties of the techniques (both analytical and statistical), unexplained signal in the range ±20% are considered good enough and not shown in the subsequent elaborations.

[Figure]

Figure R1.1. time series of PMF-factors and their unexplained mass concentrations for both AMS (upper panel) and NMR (lower panel).

Line 261 – Should the binning of the spectra also contribute to the error matrix? Averaging over the 0.02 ppm bins should include some manner of standard deviation?

Response: The error-matrix for NMR is calculated after the binning of the spectra and therefore it already represents the variability of the binned signal. Instead, the input uncertainty for NMR

dataset is calculated on purpose as independent by the signal intensity of the different peaks of the spectra (differently from AMS and from what the Referee suggests). Based on previous studies and comparative tests using different non-negative Factor Analysis techniques (Paglione et al., 2014; Paglione et al., 2024; Tagliavini et al., 2024), we believe that adding to the calculation of the input uncertainty an element related to the signal intensity is not recommended for NMR, because of the interdependence of some signals in the spectra (characterized by specific patterns and reciprocal relative intensities).

Section 3.1 – How did you define your seasons? For example, was winter based on polar night? Was spring based on polar sunrise followed by snowmelt in mid May?

Response: the seasons are separated following the general astronomical definition and so considering summer = June+July+August, fall = Sept.+Oct.+Nov., winter = Dec.+Jan.+Feb. and spring = Mar.+Apr.+May. In this way, even if winter is for sure the season most affected by polar night and spring corresponds to the polar sunrise and starting snowmelt, they are not defined based on those conditions. We added a clarification in the revised Section 3.1, as follows:

"The chemical composition of PM1 aerosol at Ny-Ålesund during the period May 2019 - June 2020 and its seasonality is summarized in Figure 2 (where summer = June + July + August, fall = September + October + November, winter = December + January + February and spring = March + April + May)."

Line 324 – Can you provide standard deviations for O:C and OM:OC?

Response: Done in the revised text.

Line 358 – Figure S14 shows correlations between timeseries of molecular tracers with the NMR factors. A similar figure for the AMS factors would be useful (e.g., IC MSA for the marine biogenic OA factor).

Response: correlations between AMS factors and main tracers (like IC MSA) are already reported in old Table S2 (becoming Table S4 in the revised version) in term of Pearson coefficients. In Fig. S14, reporting additional organic markers measured by HPLC (as explained in the main text, sect. 2.3), we decided to present the comparison with only NMR factors for two main reasons: 1- in order to reduce the number of plots and not add redundant info, considering that we already showed that AMS factors correlate well with NMR ones (see revised Table S3); 2- because the Fig. S14 mainly aims of supporting the interpretation of some signals identified within the NMR factor spectral profiles (especially in the marine POA factor, which is not distinguished by AMS.)

Line 384 – Regarding "… less methylenic long chains and a higher degree of functionalization…", are you referring to the lack of signal around 3.5 ppm? If not, do you have an explanation for that lack of signal in the marine POA factor compared to the other studies? It may help to also include references to chemical shift ranges (here and elsewhere) for the reader.

Response: as explained few lines above, the methylenic long chains are the ones accounted for by "a pattern of bands at 0.9, 1.3, 1.6, 1.8, 2.4 and 2.6 ppm of NMR chemical shift". We further added a specific info on that at the indicated line in the revised text. Moreover, we added a new Table S2 in order to help the readers in the interpretation of NMR spectral features. About the lack of signals around 3.5 ppm, it indicates a lower amount of polyols (e.g., glycerol) in the marine POA with respect to other studies, that we commented as related to a greater degree of oxidation/ageing.

Line 390 – It would be worth adding a statement that the AMS factor better correlated with the sum of the NMR factors than with either NMR factor individually (per Table S1).

Response: following the Referee' suggestion we added a statement on the fact that the sum of NMR marine POA & SOA factors corresponds quite well with the AMS Marine OA. Note that Table S1 became Table S3 in the revised version.

"The marine biogenic OA was further separated based on H-NMR data into a SOA and POA contribution (Factor 1a and 1b – F1a and F1b, respectively), the sum of which compares very well with the AMS marine OA (Figure 4 and Table S3)."

Line 409 – The comparison of factor F1b and ground types (Fig S15c) should be discussed in the main text. Currently, you discuss the role of sea ice and open ocean broadly, but your trajectory analysis provides further support for the factor identity.

Response: we thank the Referee for the suggestion. Following it we added a sentence in the main text, at the end of the discussion on F1b (L436 of the revised text), as also reported here below:

"Figure S15 (panel c) shows the ground type over which the backtrajectories of each PM1 sample were passing. In particular it highlights the higher fractional influence of sea-water vs sea-ice cover on the contributions of F1b (marine POA) as apportioned by NMR analysis, further supporting its interpretation."

Line 438 – I struggle to see a correlation between the NMR Arctic haze factor and vanillic acid / levoglucosan. To me, it appears as a comparison of noise during a time with low signal. Please provide further discussion to clarify.

Response: regarding the ancillary organic markers, considering that they were measured on a limited set of samples (covering half of the year) also characterized by different sampling resolution, our intention was to use them as confirmation of the presence/absence of possible active sources in different periods, rather than underline their correlations with the PMF factors. For this reason, we didn't report correlation coefficients between PMF-factors and ancillary organic markers in revised Table S4 (i.e., old Table S2), but we show some significant examples of time trends similarities in Figure S14. More specifically about vanillic acid, unfortunately it was measured only during a time of low concentrations, that in any case was the period of the year in which also Arctic Haze OA factor contributions were low, further confirming our interpretation. We recognize in any case that the sentence spotted out by the Referee is not completely clear/correct and so we modified it as follows:

"Thus, this OA component also correlated quite well with eBC and nss-SO4 (Table S4), and has quite low concentrations until late winter like other phenolic compounds (i.e., vanillic acid, Figure S14)."

On the other hand, correlation with levoglucosan is better discussed and clarified in revised Table S4 and Fig. S17, based on levoglucosan quantifications from NMR (i.e., on the same samples and for the whole year-long time period of the PMF-factors)

Line 448 – The reference to Fig. S14 seems out of place since you discuss the AMS factor while Fig. S14 shows the NMR factors.

Response: true, thanks for noticing the wrong reference. We corrected it referring now to Fig. S7 (showing in details the AMS factor profiles and elemental ratios).

Line 463 – The brief explanation of the multilinear regression is hard to follow. Assuming I understand correctly, I suggest rephrasing in a manner similar to "Total OC mass was fit using a linear combination of the WSOC PMF factors (using AMS and NMR factors independently). The multiplicative coefficients are considered to be recovery coefficients (*RC*), which are inversely related to solubility."

Response: we thank the Referee for his/her suggestion. We rephrased the sentence accordingly.

Line 465 – Following from the previous comment, the reader would benefit from a simple explanation on how to interpret the fitting coefficients. For example: "higher coefficients mean the corresponding factor was less water soluble, and is associated with a higher fraction of insoluble OC." Again, assuming my understanding is correct.

Response: we agree with the Referee and we added this explanation in the revised text.

Line 503 – Table 1 should be mentioned earlier and discussed in the previous paragraphs. When discussing each factor, mention the average mass contributions during relevant seasons (e.g., line 487 marine OA in summer (11% and 14% in AMS and NMR, respectively) and spring (9% and 10%)).

Response: we accept the Referee' suggestion and we mentioned Table 1 earlier (together with Figure 6). We also added the numeric value of the average mass contributions (seasonal or annual) of each factor when discussed in the text.

Fig. 4 – Are the NMR F1a and F1b factor timeseries stacked on top of one another? If so, it would be helpful to note that in the caption.

Response: yes, of course, F1b is stacked on top of F1a in order to highlight that their sum resembles quite well the AMS marine OA. We added a note in the caption of the revised version.

Table 1 – I suggest reformatting for clarity. At first, I thought this table was trying to show the variability of these factors within a particular season. I suggest adding two columns per factor, one for AMS and one for NMR to more clearly show that this table is meant to compare factors between the two instruments. With that said, does this table not simply repeat information in Fig. 6? To help this table add information, you should also include the standard deviations (i.e., the variability within each season and annually).

Response: following the Referee' suggestion we modified Table 1 separating AMS and NMR results and adding standard deviations, as follows:

| Ave ±std.dev | marine SOA | | marine POA | | Aged wildfires OA | | Artic haze OA | | Background OA | |
|---|---|---|---|---|---|---|---|---|---|---|
| | AMS | NMR | AMS | NMR | AMS | NMR | AMS | NMR | AMS | NMR |
| SUMMER | 11 ±11% | 14 ±11% | 20 ±15% | 30 ±20% | 28 ±21% | 27 ±22% | 7 ±8% | 6 ±6% | 34 ±25% | 23 ±18% |
| FALL | 1.9 ±3.1% | 2.1 ±1.7% | 1±2% | 7 ±8% | 25 ±10% | 17 ±16% | 24 ±24% | 31 ±15% | 48 ±21% | 43 ±19% |
| WINTER | 0.3 ±0.8% | 0.7 ±0.5% | 1 ±3.3% | 3 ±3% | 11 ±6% | 7 ±8% | 53 ±21% | 70 ±9% | 35 ±20% | 19 ±12% |
| SPRING | 9 ±15% | 10 ±13% | 2 ±3% | 7 ±5% | 14 ±9% | 33 ±17% | 31 ±23% | 27 ±20% | 44 ±16% | 23 ±17% |
| Whole | 5 ±10% | 6 ±10% | 7 ±12% | 12 ±16% | 20 ±15% | 21 ±19% | 28 ±26% | 33 ±27% | 40 ±22% | 28 ±19% |

Then, following also the Editorial suggestion, we removed the colors from the cells.

SI Fig. S2 – What kind of regression is being used (least-squares, orthogonal distance, etc.)? Is it weighted by measurement uncertainties? Similar for regressions in Fig. S3.

Response: The linear fits in Fig. S2 and S3 used the ordinary least squares (OLS) regression. They are not weighted for measurement uncertainties.

SI Fig. S4 – Include a legend.

Response: Done, thanks for suggesting.

Fig. S5a – For clarity, start the y axis at 0.5.

Response: done.

SI Lines 70 – For clarity, provide the downweighting factor.

Response: done. We modified the sentence as follows:

"All fragments with a signal-to-noise ratio (SNR) below 0.2 were removed from the matrices, and those with a signal-to-noise ratio below 2.0 were down-weighted, according to the recommendations of Paatero and Hopke (2003), increasing their uncertainty by a factor 2."

SI Fig S14 – The plots comparing NMR factor 4 with various tracers all show a reduction in those tracers around August and September. Factor 4 doesn't show a similar reduction. Please include discussion on (i) why these tracers might be lower during this time frame, and (ii) your thoughts on why factor 4 isn't also lower during that time frame.

Response: we thank the Referee for the careful examination. The particular period spotted out is a nice example of the great deal of variability observed in the time trend of NMR factor 4 at the weekly-to-submonthly time scales, especially in spring and summer. Such variability must be put in relation to changes in air masses at synoptic spatio-temporal scales. Investigating such source of variability on a case-study basis is beyond the scope of the present study, while we provide here a synthetic analysis of the effect of atmospheric transport at synoptic scale on the concentrations of the individual organic factors using the CWT approach. While in the present paper we discuss the main temporal patterns (e.g. seasonality), a more tailored analysis of specific episodes can be object of future investigations. About the specific period of Sept 2019, the Referee correctly noticed that the time trend of NMR factor 4 tended to diverge from those of many organic tracers considered in Fig. S14. This is not completely true, however, for the concentrations of polyols (sorbitol and, to some extent, mannitol) showing a ramp-up in Sept. Additional species that can contribute to the sustained concentrations of NMR factor 4 in Sept 2019 may include acetic and glycolic acids which, however, were not considered in Fig. S14 because of the too few valid data available. Nevertheless, we agree with the Referee that the trends of the organic tracers encompassing the polyols, organic acids and aminoacids used as ancillary data in our study cannot fully reconstruct the complete trend of NMR factor 4, whose chemical nature is probably originating from diverse chemical classes, probably including compounds unaccounted for by the available molecular speciation.

SI Fig. S16 – Could you add some annotations for key peaks, similar to Fig. S13?

Response: Done.

SI Fig. S16 – There appears to be signal ~6.5-8.5 ppm in the "standards" and EUCAARI factor analysis. Is there a proposed reason why this signal is absent in this study's wildfire factor?

Response: The Referee is correct, the aromatic groups detectable by H-NMR spectroscopy between 6.5 and 8.5 ppm of chemical shift characterize our reference spectra for biomass burning aerosols to a very variable extent and are completely missing in the spectrum of the NMR factor for aged

wildfire smoke in Ny-Ålesund. As aromatic compounds can be transformed or degraded but not produced by atmospheric ageing, the extent of ageing reactions (beside the effect of the variability at the source) affects the amount of aromatic biomass burning compounds at the receptor site. With respect to the reference spectra shown in Fig S16, the Ny-Ålesund NMR aged wildfire smoke spectrum is therefore characterized by extensive ageing.

SI Line 250 – The factors' contributions to the total OA "…varied by less than 30 %..." (SI line 251). These uncertainties should be acknowledged in the main text and used in the discussion of comparing the AMS and NMR factor contributions (e.g., line 505). The presented factor (Fig. 4) does not seem to be the average presented in the SI (Fig. S19). How does the presented factor compare to the bootstrapping average?

Response: the reported 30% is considered as a general acceptable range of uncertainty for PMF results and is actually the max value of only one sample for one factor (AMS F3, sample 18-Nov-2019), while the average uncertainties are much lower (in the range 3-7% for AMS factors and 2-9% for the NMR ones). Max values (meaning samples with the highest uncertainty for each factor) range is 8-30% for AMS and 6-28% for NMR factors.

About Fig. S19, the rather large error bars (especially the one associated to the sample 04-jul-19) distort the depiction of the time trend of NMR F4 (which in any case shows the corrected bootstrapping average). Removing error bars, the similarities are much more evident, as we show here below for the Referee, comparing F4 time trend from Fig. 4 with the bootstrapping average (Fig. R1.2).

[Figure]

Figure R1.2. comparison between time series of the F4 contributions from the chosen NMR-PMF solution (dark yellow shaded area) and from the bootstrapping average (orange dotted line and markers).

SI Line 251 – Should the reference to factors' averages and standard deviations be Fig. S19, not S17?

Response: right, thanks. We changed the reference.

SI Line 301 – The claim that using ugC and umolH yielded similar results needs further discussion. Be quantitative in how they are similar. Perhaps normalized ratios of the fitting parameters? Or comparisons of the reconstructed total OC?

Response: following the Referee' suggestion, we added a new supplementary figure (Figure S20) showing the comparison between results of the MLR model applied both to the NMR and AMS factors, expressed both in term of μgC m-3 or alternatively in term of μmolH m-3.

[Figure]

**Figure S20.** Comparison between results of multilinear regression model applied both to the NMR and AMS factors (left and right side, respectively), expressed in term of μgC m$^{-3}$ (upper charts) or alternatively in term of μmolH m$^{-3}$ (lower charts). Pie charts report the relative contributions of the PMF-factors as annual averages.

Table S4 – I'm not sure where "beta" is used? Is it meant to refer to a value in equation S4?

Response: right, actually what is called "beta" in old Table S4 (becoming Table S6 in the revised version) is the Recover Coefficient "RC" reported in Eq. S4. We corrected the notation, thanks.

**Technical Comments**

Line 49-50 – Two instances of "increased."

Response: corrected, thanks.

Line 63 – EC has not yet been defined.

Response: done, thanks.

Lines 71 and 74 – I think references to Moschos et al. (2022b) should be just (2022) since there are no references to other studies by Moschos et al. that I can see.

Response: right, thanks.

Line 112 – TOC should be defined.

Response: done.

Line 213 – Define OM.

Response: done, thanks.

Line 307 – GVB is not defined.

Response: now is defined in the previous Section 2.1.

Line 355 – "…(for details see Supplementary Section S2, Figure S6-S18)." For clarity, I recommend moving this portion of the statement up to line 353 (maybe insert alongside "4 for AMS, 5 for NMR"). Section S2 does not discuss the agreement between the techniques, and instead is a more general discussion of the PMF analysis.

Response: we accepted the Referee' suggestion and moved the statement accordingly.

Line 442 – Ny-Ålesund is missing a hyphen.

Response: added, thanks.

Line 453 – VOC is not defined.

Response: done.

Line 504 – "With" respect to?

Response: corrected, thanks.

Line 516 – "…summertime OA resulted the less oxidized…" Should this say something like "summertime OA was less oxidized"?

Response: yes, we rephrased it as suggested.

Fig. 2a, 3a – The -3 in the y axis label should be superscript.

Response: done.

Figs. 2, 4, 5 - Ny-Ålesund is missing the hyphen and accent on the Å in the captions.

Response: corrected, thanks.

SI Fig. S2 – Panels b and c use AMS HROrg while the caption uses AMS WSOM.

Response: we replaced HROrg with WSOM.

SI Fig. S7 – caption says 5 factors while the figure shows 4.

Response. Corrected, thanks.

---

## Author Comment (AC2)

We thank the Reviewer #2 for the helpful and constructive comments, which have led to significant improvements in the manuscript. We have carefully revised the text. Our point-by-point replies are given below (blue), following the referees' comments (black). Changes to the manuscript are marked with green.

**Reviewer 2**:

**General comments**

This paper presents a one-year data set of PM1 samples obtained in Ny-Ålesund with the bulk chemical analytical results of H-NMR spectroscopy and off-line HR-ToF-AMS. The authors show that the analytical results of the two measurements are consistent in terms of source contributions to OAs. Their data suggested that the observed OAs in winter and spring are dominated by long-range transport of anthropogenic pollution in Eurasia, while the aerosol in summer is characterized by biogenic aerosols from marine sources. Overall, the paper provides new insights into our understanding of the seasonality in the source contributions to OAs in the Arctic region and confirms some of the findings already reported in previous studies. While the data presented are valuable and interesting, there are some issues that need to be clarified before its publication in ACP.

Response: We appreciate the general Reviewer's positive feedback on the manuscript relevance.

**Specific comments**

(1) From the text, the contributions of terrestrial BSOA traced by oxidation products of terpenes are not clear. The authors attributed these BSOAs to emissions from wildfires but this is not always the case. Indeed, Moschos et al (2022) reported that significant or non-negligible amounts of BSOA (not necessarily related to biomass burning) from forests for the observed OA in pan-Arctic in summer. I think that the authors should add more discussions on this point (e.g., if the author's result is different from Moschos et al., why?).

Response: we thank the Referee for the careful examination and suggestion. We decided to attribute wildfires as main source for F2 because of a combination of its spectral features (HULIS features for NMR, many oxygenated fragments at high m/z for AMS), resembling very aged continental biomass burning emissions, and its temporal/geographical pattern (sporadic very high concentrations corresponding to fire events). But we wanted also to spot out that in NMR spectra of specific samples associated to F2, we see signals related to BSOA for which in any case the PMF was not able to separate a specific factor in NMR nor in AMS. So, we cannot separate and quantify a specific BSOA contribution within our dataset. We interpreted it as the result of a strong co-variation between the BSOA signals and the others signals of F2 more characteristics of aged BB. We made the hypothesis that what we sporadically traced at the receptor site of GVB were the BSOA co-emitted during fire events (characterized by higher temperature/convection facilitating the subsequent transport to long distances). The alternative hypothesis (also considered plausible and reported in the main text) is that BSOA signals identified in our dataset represent oxidation products of forest emissions (terpene and isoprene) moving to Ny-Alesund from the same area (boreal forests in Eurasia), contributing to a variable fraction of our F2 but also to F4 (Background mix) that shows some biogenic terrestrial signals as well and has a slight increase during polar-day months (Fig. 4).

Given these considerations, to answer the Referee, we add that:

1) the overall picture that emerges from our hypotheses and descriptions is not actually inconsistent with that of Moschos et al. (2022): in Moschos during the polar day for both Gruvebadet (9%) and Zeppelin (19%), bioSOA is a minor contributor to OC in PM10. We believe that in Moschos' study the Authors managed to separate such a specific BSOA factor because there were included stations from the continents (e.g. Pallas), where bioSOA is a major contributor. Whether the attributed concentrations (9-19%) are that reliable in Svalbard or not would probably need more future confirmations. In any case the picture emerging in our study is consistent with a portion of 9-19% of BSOA included partially in F2 and/or in F4 (representing up to 28% and up to 34% of organic PM1, for F2 and F4 respectively, as summertime averages).

2) the time series, air mass origin and spectral features of F2 show pretty clearly that aged fires are the main source attributable to this factor, supporting our interpretation. Compounds identical/similar to bioSOA can also be emitted during fires and we very likely traced them in F2.

3) we agree with the Referee that we should explicitly refer to Moschos et al. findings in our discussion and we should clarify better the possibility that BSOA can contribute partially also to F4. Aiming these objectives, we added few sentences in Section 3.3 and Section 4.

(2) P.7, L.215: WSOM = WSOC×(OM:OC)_AMS

I understand that the advantage of the use of OM:OC at the time of each sampling is to be able to expect more realistic abundance of WSOM rather than by use of a constant value of the factor. Meanwhile, the composition of OM minus OC can also include water-insoluble compound mass. How can this use of OM:OC ratio be verified (or is there any evidence) to represent water-soluble mass? The authors should add some more description including uncertainty in this calculation.

Response: given that AMS analyses were done off-line on water-extracts of the filters (as clearly explained in Section 2.2.2), the OM:OC ratios refer to the water-soluble fraction of OA by definition. We do not believe it is necessary to add more explanations/evidences of that in the text.

Nonetheless, an evidence of the consistency of OM:OC ratios is the fact that if we apply OM:OC ratios to the WSOM mass measured by AMS (i.e., WSOC_AMS = WSOM_AMS / OM:OC_AMS), the resulting WSOC_AMS corresponds quite well with the WSOC measured by TOC-analyzer (as shown in the following plot for the Referee)

[Figure]

(3) P.10, L.295: "The major chemical mass …, followed by.."

I think that this statement may cause misunderstandings of readers: some may think that sulfate was the most abundant and the second most is seasalt, followed by OM. However, there seems to be no statistical difference in the fraction among sulfate, seasalt, and OM. As the authors described in the conclusion section, these three components had similar contributions to the PM1 mass. Please modify the sentence.

Response: we thank the Referee for the suggestion. We rephrased the sentence accordingly, as follows:

"On yearly average the PM1 was mainly constituted in similar proportions of nss-sulfate (representing 33 ± 13% of the total), seasalt (29 ± 13%) and OM (28 ± 16%, of which 22 ± 14% represented by WSOM), the rest being accounted for by much smaller contributions of ammonium (4 ± 2%), nitrate (1 ± 1%), eBC (2 ± 2%) and other non-sea salt ions (i.e., nss-K, nss-Mg and nss-Ca, amounting to 3 ± 4% in total)."

(4) P.11, after L. 329: Regarding the statement staring with "MSA and …," which figure is referred to? Maybe Fig. 4? Please clarify it.

Response: no, it is referred to Figures 3 and S4, as mentioned on the line above. We added in the revised version more references to the proper Figures along the description.

(5) Figure 5: This figure is very hard to see. For example, the color code is not clear what it represents. The authors explain it in the caption with quantitative information, but they should show the color code in the figure panel in addition to describing it in the caption. Moreover, geographical lines (map) in the figure are not clear at all.

Response: we thank the Referee for the suggestion. We modified the Figure accordingly, adding the color scale to each map and changing colors of the continental edges lines in order to improve readability.

(6) Figure 6 and P.16: As the author described, the scaled contributions of the NMR and AMS factors to total OC showed generally good agreement. However, the relative contributions of background OA, Arctic haze OA, and Aged wildfires OA between NMR and AMS particularly in spring are significantly different. The authors should add more discussion on the possible reason for this difference.

Response: we already openly acknowledged in the text the discrepancies and attributed them mainly to the different sensitivity of the two techniques to specific classes of organics present in the complex mixtures of compounds that constitute OA; for instance, NMR seems to be more sensitive to fatty acid chains attributing more mass to marine POA (see Fig.4), while it "misses" something of the aged continental BBOA and the Arctic haze compared to AMS, which are enriched of branched polysubstituted oxygenated species. This depends on the fact that actually H-NMR is measuring directly the resonances of hydrogen atoms (-H) of functional groups (meaning specific kind of bonds and molecular structures) bearing not-exchangeable protons, like aliphatic -CH2 chains of fatty acids, but cannot quantify directly carbon atoms not protonated (or bearing exchangeable protons), like aliphatic carbonyl/carboxyl groups. AMS on the other hand is really sensitive to the latter (and less to the former). However, this level of detailed discussion is extremely technical and would remain basically speculative without extensive laboratory tests which, honestly, go beyond the scope of the present manuscript. For this reason, we prefer not to address this with an extensive additional discussion; instead, we modified the text to highlight that evidencing such discrepancies in the OA source apportionment is a further important finding emerging from the present study, which provides a measure of the level of bias one may encounter when relying on a single technique to characterize OA.

We changed the last part of Section 4 as follows:

"Although the AMS and NMR showed an overall good agreement in OC source apportionment, some discrepancies could be noticed in the relative contributions of specific components to the aerosol OC (Table 1). While the total marine and the continental aged BB & BSOA fractions agreed quite well, a greater contribution for Factor 4 (background OA) respect to Factor 3 (Arctic haze OA) was derived by AMS when comparing to NMR (for background OA 40 versus 28% and for Arctic haze OA 28 versus 33% on yearly average, for AMS and NMR respectively). We believe that such discrepancies, likely related to the different sensitivity of the two instruments to specific organic mixtures (see Sect. 2.2.1 and 2.2.2), provide a measure of the level of bias one may encounter when relying on a single technique to characterize OA, representing a further relevant output of this study. And despite these discrepancies, the overall agreement between NMR and AMS characterizations highlights the robustness of the study's findings and reveals a consistent picture of the main organic submicron aerosol sources in Ny-Ålesund and their seasonality."